# Therapeutic effects of anodal transcranial direct current stimulation in a rat model of ADHD

Da Hee Jung[1,2], Sung Min Ahn[3], Malk Eun Pak[3], Hong Ju Lee[1,2], Young Jin Jung[4], Ki Bong Kim[5], Yong-Il Shin[6], Hwa Kyoung Shin[1,2,3], Byung Tae Choi[1,2,3]*

[1]Department of Korean Medical Science, School of Korean Medicine, Pusan National University, Yangsan, Republic of Korea; [2]Graduate Training Program of Korean Medicine for Healthy Aging, Pusan National University, Yangsan, Republic of Korea; [3]Korean Medical Science Research Center for Healthy Aging, Pusan National University, Yangsan, Republic of Korea; [4]Department of Radiological Science, Health Science Division, Dongseo University, Busan, Republic of Korea; [5]Department of Korean Pediatrics, School of Korean Medicine, Pusan National University, Yangsan, Republic of Korea; [6]Department of Rehabilitation Medicine, School of Medicine, Pusan National University, Yangsan, Republic of Korea

**\*For correspondence:**
choibt@pusan.ac.kr

**Competing interests:** The authors declare that no competing interests exist.

**Abstract** Most therapeutic candidates for treating attention-deficit hyperactivity disorder (ADHD) have focused on modulating the dopaminergic neurotransmission system with neurotrophic factors. Regulation of this system by transcranial direct current stimulation (tDCS) could contribute to the recovery of cognitive symptoms observed in patients with ADHD. Here, male spontaneously hypertensive rats (SHR) were subjected to consecutive high-definition tDCS (HD-tDCS) (20 min, 50 μA, current density 63.7 A/m$^2$, charge density 76.4 kC/m$^2$) over the prefrontal cortex. This treatment alleviated cognitive deficits, with an increase in tyrosine hydroxylase and vesicular monoamine transporter two and significantly decreased plasma membrane reuptake transporter (DAT). HD-tDCS application increased the expression of several neurotrophic factors, particularly brain-derived neurotrophic factor (BDNF), and activated hippocampal neurogenesis. Our results suggest that anodal HD-tDCS over the prefrontal cortex may ameliorate cognitive dysfunction via regulation of DAT and BDNF in the mesocorticolimbic dopaminergic pathways, and therefore represents a potential adjuvant therapy for ADHD.

## Introduction

Attention-deficit hyperactivity disorder (ADHD) is a heterogeneous neuropsychiatric disorder highly prevalent in children, characterized by impairments in attention and/or hyperactivity-impulsivity (*Faraone et al., 2015*; *Tsai, 2017*). Psychostimulants, such as amphetamine and methylphenidate (MPH) that work by increasing central dopamine and norepinephrine activity in the brain, are recommended as first-line pharmacological therapy for patients with ADHD (*Faraone, 2018*). Although these drugs impact executive and attentional functions, some patients fail to respond or experience adverse effects including cardiovascular effects, and discontinue the treatment (*Faraone, 2018*; *Tsai, 2017*).

The pathogenesis of ADHD is not fully understood, but genetic factors play a significant role in its development (*Kent, 2004*). Most therapeutic candidates have focused on modulating the dopaminergic neurotransmission system, with additional candidates involving the noradrenergic and serotonergic systems (*Banaschewski et al., 2010*; *Kent, 2004*). In particular, the neurotransmitter

dopamine plays an important role in behavioral processes such as cognition and emotional processing; therefore, its dysfunction is involved in several neuropsychiatric disorders including ADHD (*Faraone and Biederman, 1998*; *Leo et al., 2018*). Alterations in dopaminergic neurotransmission within the mesocorticolimbic system are involved in the pathophysiology of ADHD, with functional abnormalities in fronto-basal ganglia networks (*Biederman and Faraone, 2002*; *Hart et al., 2013*).

Genes associated with neuronal development and plasticity are considered another important target in the clinical manifestation and pathogenesis of ADHD (*Banaschewski et al., 2010*; *Galvez-Contreras et al., 2017*; *Tsai, 2017*). Neurotrophic factors (NTFs) are essential for neural development of the brain and plasticity in adults and are involved in the pathogenesis of ADHD (*Bilgiç et al., 2017*; *Tsai, 2017*). In particular, brain-derived neurotrophic factor (BDNF) has been identified in the pathophysiology of ADHD and represents a biological target for treatments of this disorder (*Conner et al., 2008*; *Kent et al., 2005*; *Tsai, 2017*).

Transcranial direct current stimulation (tDCS) has been extensively investigated in children with a variety of diagnoses, including neuropsychiatric disorders, and has shown no serious adverse effects (*Bikson et al., 2016*). tDCS improves short- and long-term memory deficits and is associated with altered dopamine levels and enhanced synaptic activity, respectively, as shown in an ADHD animal model (*Leffa et al., 2018*; *Leffa et al., 2016*). Anodal tDCS exerts beneficial effects on higher order cognitive functions, such as working memory, attention, and perception, in patients with diabetes and animal models through the augmentation of synaptic plasticity, thus requiring BDNF secretion (*Coffman et al., 2014*; *Wu et al., 2017*). Because ADHD has comorbid cognitive dysfunction disorders (*Faraone et al., 2015*; *Tsai, 2017*), tDCS has been proposed as a possible therapeutic option for treating patients with ADHD (*Leffa et al., 2018*; *Leffa et al., 2016*).

Conventional tDCS using saline sponge-based rectangular pads stimulates a large scalp area; thus, the current flow is not concentrated on targeted neuronal populations (*Brunoni et al., 2012*). However, high-definition tDCS (HD-tDCS) uses small ring-based electrodes to facilitate stimulation and target current delivery, thereby overcoming the lack of specificity (*Datta et al., 2009*). As HD-tDCS is a potential tool for brain stimulation in the treatment of cognitive deficits, we hypothesized that the regulation of dopaminergic neurotransmission and NTFs using an HD-tDCS approach would contribute to the recovery from cognitive symptoms in patients with ADHD. We therefore modified anodal HD-tDCS for rodents using a small ring-based electrode to optimize focality and intensity. The aim of this study was to identify the therapeutic effects of modified HD-tDCS in a preclinical model of ADHD and to investigate the underlying mechanisms related to these effects. We evaluated the therapeutic effects of HD-tDCS using behavioral assessments of cognitive functions, and performed biochemical and immunofluorescence assays to investigate alterations in dopaminergic neurotransmission and NTFs at the core sites of dopaminergic pathways.

## Results

### Effects of HD-tDCS application on the cognitive dysfunctions in the ADHD rat model

We performed various cognitive tests to investigate whether HD-tDCS application improves functions in this rat model of ADHD. In the open-field test, the spontaneously hypertensive (SHR) and sham animals showed significant hyperactivity compared to the Wistar-Kyoto rat (WKY) group, which was reduced by HD-tDCS and MPH treatment, especially in the tDCS-PFC group (*Figure 1A*, $F_{(5,42)}$=5.984, p<0.001 and *Figure 1—figure supplement 1A*). In the delayed non-match to place (DNMTP) version of the T-maze, HD-tDCS and MPH treatment significantly increased performance. Moreover, the tDCS-PFC group showed more improvement than the tDCS-M1 group (*Figure 1B*, $F_{(5,54)}$=116.861, p<0.001 and *Figure 1—figure supplement 1B*). In the Y-maze, the tDCS-M1 group showed a significant decrease in time spent in the new arm compared to the SHR group in the modified version; however, the tDCS-PFC and MPH groups showed significantly increased spontaneous alternation compared to the sham group in the alternation task, especially with respect to same-arm returns (SAR) (*Figure 1C and D*, $F_{(5,36)}$=0.750, p=0.591 and *Figure 1—figure supplement 1C and D*). Retention latency in the passive avoidance test was significantly increased in the HD-tDCS and MPH-treated groups compared to the sham group (*Figure 1E*, $F_{(5,30)}$=31.245, p<0.001). In the object-place recognition test, all groups demonstrated a similar total distance during training and

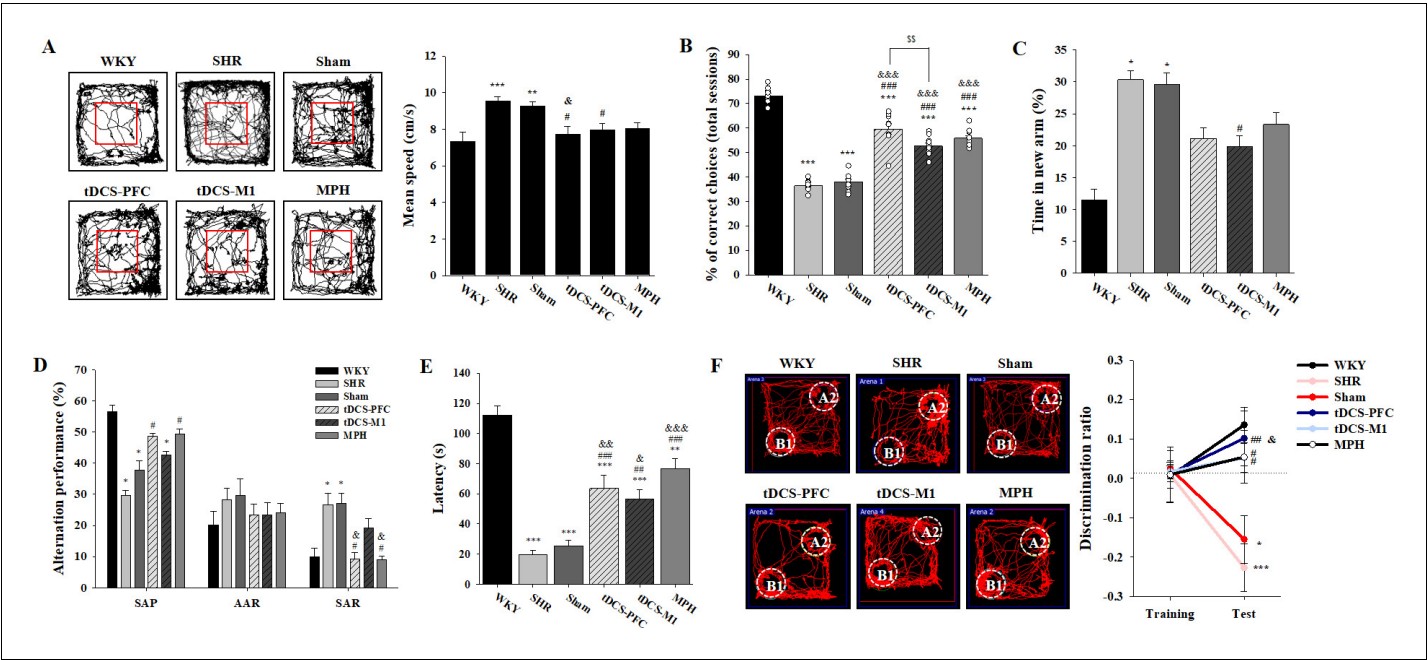

**Figure 1.** Effects of HD-tDCS application on cognitive behaviors in a rat model of ADHD. (**A**) Performance and mean speed on the open-field test. Hyperactivity in the open-field test was significantly reduced in the tDCS-PFC group compared with the sham group (n = 8). (**B**) A DNMTP of the T-maze results, total % of correct choices for 10 sessions. The % correct choices were significantly higher in all HD-tDCS and MPH-treated groups compared to the sham group (n = 7). (**C**) Modified Y-maze results, % of time in new arm. The time in the new arm was significantly decreased in the tDCS-M1 group compared with the SHR group (n = 8). (**D**) Y-maze spontaneous alternation task, % alternation performance of SAP, AAR, and SAR. SAR was significantly lower in the tDCS-PFC and MPH groups compared with the sham group (n = 7). (**E**) Passive avoidance test, latency. HD-tDCS and MPH-treated groups showed significantly higher latency compared to the sham group (n = 6). (**F**) Object-place recognition test, representative examples of movement path and discrimination ratio. The tDCS-PFC group showed a marked increase in time spent exploring the novel placed object compared to the sham group (n = 8). SAP; spontaneous alternation performance, AAR; alternate arm return. Data are presented as mean ± SEM. *$p<0.05$, **$p<0.01$, and ***$p<0.001$ vs. WKY; #$p<0.05$, ##$p<0.01$, and ###$p<0.001$ vs. SHR; &$p<0.05$, &&$p<0.01$, and &&&$p<0.001$ vs. sham; $$$p<0.01$ vs. tDCS-PFC.

The online version of this article includes the following source data and figure supplement(s) for figure 1:

**Source data 1.** Source files for behavior tests.
**Figure supplement 1.** Additional data on cognitive behavior following HD-tDCS application in our rat model of ADHD.
**Figure supplement 2.** Effects of HD-tDCS application on cognitive behaviors in WKY rats, the genetic control model of ADHD.

test sessions. The discrimination ratio was significantly increased in the HD-tDCS and MPH-treated groups compared to the SHR group; however, only the tDCS-PFC group showed significant changes compared to the sham results (*Figure 1F*, $F_{(5,42)}=6.554$, $p<0.001$ and *Figure 1—figure supplement 1E*). To confirm that WKY are the ideal control for SHR rats, we performed a behavioral analysis on hyperactivity and cognitive performance, comparing the WKY, WKY-sham, and WKY-tDCS-PFC groups. There was no significant difference among these groups in the open-field test and the DNMTP (*Figure 1—figure supplement 2*). These results suggest that HD-tDCS application alleviates cognitive dysfunction in this ADHD rat model, especially after stimulation over the prefrontal cortex.

## Effects of HD-tDCS application on gene expression of dopaminergic neurotransmission factors and NTFs in the ADHD rat model

Next, we compared the gene expression of dopaminergic neurotransmission factors such as tyrosine hydroxylase (TH), plasma membrane reuptake transporter (DAT), and vesicular monoamine transporter 2 (VMAT2) as well as NTFs including BDNF, transforming growth factor-beta1 (TGF-β1), glial-cell-derived neurotrophic factor (GDNF), nerve growth factor (NGF), and neurotrophin-3 (NT3) in the core regions of dopaminergic projections, that is, the prefrontal cortex, striatum, hippocampus, and substantia nigra/ventral tegmental area (SN/VTA). Generally, the HD-tDCS and MPH-treated groups showed increased *TH* and *VMAT2* gene expression compared to the sham group, whereas *DAT*

gene expression was decreased. These patterns were especially pronounced in the hippocampus and the SN/VTA region. Compared to the sham group, the tDCS-PFC group showed a significant increase in *TH* and *VMAT2* gene expression in the SN/VTA and prefrontal cortex, respectively, and a significant decrease in *DAT* gene expression in all other regions except the prefrontal cortex. A significant decrease in *DAT* expression was also observed in the hippocampus and SN/VTA region of the tDCS-M1 group. However, the MPH group showed significant changes in *TH* and *DAT* expression only in the striatum (*Table 1*).

As for NTF gene expression, an increase in expression in the HD-tDCS and MPH treatment groups was observed, especially for the *BDNF* gene. Increases in NTF genes were most noticeable in the hippocampus, followed by the SN/VTA. Compared to the sham group, the tDCS-PFC group showed a significant increase in *BDNF* gene expression in the hippocampus, increased *TGF-β1* expression in the striatum and SN/VTA, increased *GDNF* expression in the prefrontal cortex and hippocampus, increased *NGF* expression in the hippocampus, and increased *NT3* expression in the prefrontal cortex and striatum. The tDCS-M1 group showed significantly increased *BDNF* expression in

**Table 1.** Effect of HD-tDCS on gene expressions of dopaminergic neurotransmission factors in the prefrontal cortex, striatum, hippocampus, and SN/VTA (n = 4).

| Fold change (of WKY) | SHR | Sham | tDCS-PFC | tDCS-M1 | MPH | Sham vs. tDCS-PFC | Sham vs. tDCS-M1 | Sham vs. MPH | PFC vs. M1 |
|---|---|---|---|---|---|---|---|---|---|
| **Prefrontal cortex** | | | | | | | | | |
| *TH* | 0.29 ± 0.05 | 0.31 ± 0.05 | 0.34 ± 0.04 | 0.26 ± 0.17 | 0.24 ± 0.05 | p=0.493 | p=0.686 | p=0.078 | p=0.486 |
| *DAT* | 0.34 ± 0.24 | 0.51 ± 0.39 | 0.76 ± 0.69 | 0.50 ± 0.45 | 0.59 ± 0.38 | p=0.552 | p=0.963 | p=0.783 | p=0.544 |
| | | | | | | F(1,6)=0.397 | F(1,6)=0.00237 | F(1,6)=0.0828 | F(1,6)=0.413 |
| *VMAT2* | 0.19 ± 0.03 | 0.13 ± 0.08 | 0.31 ± 0.11 | 0.23 ± 0.18 | 0.20 ± 0.02 | **p=0.040** | p=0.355 | p=0.343 | p=0.475 |
| | | | | | | **F(1,6)=6.800** | F(1,6)=1.002 | | F(1,6)=0.579 |
| **Striatum** | | | | | | | | | |
| *TH* | 0.50 ± 0.16 | 0.67 ± 0.21 | 0.63 ± 0.10 | 1.14 ± 0.33 | 1.37 ± 0.24 | p=0.778 | p=0.054 | **p=0.005** | **p=0.027** |
| | | | | | | F(1,6)=0.0868 | F(1,6)=5.724 | **F(1,6)=19.243** | **F(1,6)=8.493** |
| *DAT* | 1.38 ± 0.58 | 1.32 ± 0.23 | 0.41 ± 0.17 | 1.01 ± 0.34 | 0.77 ± 0.14 | **p=<0.001** | p=0.176 | **p=0.006** | **p=0.029** |
| | | | | | | **F(1,6)=41.344** | F(1,6)=2.349 | **F(1,6)=17.006** | |
| *VMAT2* | 0.33 ± 0.25 | 0.63 ± 0.27 | 0.36 ± 0.17 | 1.00 ± 0.80 | 0.67 ± 0.41 | p=0.140 | p=0.886 | p=0.870 | p=0.343 |
| | | | | | | F(1,6)=2.891 | | F(1,6)=0.0292 | |
| **Hippocampus** | | | | | | | | | |
| *TH* | 0.47 ± 0.15 | 0.45 ± 0.21 | 0.80 ± 0.28 | 0.62 ± 0.17 | 0.44 ± 0.06 | p=0.091 | p=0.242 | p=1.000 | p=0.320 |
| | | | | | | F(1,6)=4.061 | F(1,6)=1.683 | | F(1,6)=1.173 |
| *DAT* | 1.42 ± 0.44 | 1.48 ± 0.38 | 0.60 ± 0.16 | 0.92 ± 0.17 | 1.16 ± 0.60 | **p=0.005** | **p=0.037** | p=0.397 | **p=0.031** |
| | | | | | | **F(1,6)=18.468** | **F(1,6)=7.101** | F(1,6)=0.831 | **F(1,6)=7.884** |
| *VMAT2* | 0.51 ± 0.18 | 0.56 ± 0.27 | 0.92 ± 0.34 | 0.79 ± 0.25 | 0.48 ± 0.08 | p=0.147 | p=0.246 | p=0.597 | p=0.574 |
| | | | | | | F(1,6)=2.776 | F(1,6)=1.650 | F(1,6)=0.311 | F(1,6)=0.353 |
| **SN/VTA** | | | | | | | | | |
| *TH* | 0.32 ± 0.04 | 0.30 ± 0.18 | 1.09 ± 0.57 | 0.46 ± 0.28 | 0.65 ± 0.41 | **p=0.039** | p=0.343 | p=0.343 | p=0.094 |
| | | | | | | **F(1,6)=6.880** | | | F(1,6)=3.947 |
| *DAT* | 1.20 ± 0.20 | 1.62 ± 0.55 | 0.58 ± 0.18 | 0.85 ± 0.13 | 1.02 ± 0.47 | **p=0.011** | **p=0.034** | p=0.146 | p=0.056 |
| | | | | | | **F(1,6)=12.964** | **F(1,6)=7.526** | F(1,6)=2.781 | F(1,6)=5.618 |
| *VMAT2* | 0.22 ± 0.08 | 0.49 ± 0.27 | 0.92 ± 0.53 | 1.00 ± 0.47 | 0.89 ± 0.46 | p=0.343 | p=0.111 | p=0.175 | p=0.843 |
| | | | | | | | F(1,6)=3.499 | F(1,6)=2.362 | F(1,6)=0.0428 |

Expression of the *TH, DAT,* and *VMAT2* genes, expressed as fold changes of WKY. Data represent the mean ± SEM. Data were analyzed using ANOVA for repeated measures, followed by Tukey's tests for multiple comparisons. p<0.05 was considered statistically significant; significant results are highlighted in bold.

the prefrontal cortex and hippocampus and *NT3* expression in the striatum. Moreover, all NTF genes examined (except *GDNF*) were significantly upregulated in this group. The MPH group showed significant changes in *BDNF* and *GDNF* gene expression in the striatum, and in *NGF* and *NT3* expression in the hippocampus (*Table 2*). These results suggest that HD-tDCS application regulates the gene expression of dopaminergic neurotransmission factors and NTFs, in particular the *DAT* and the *BDNF* gene, in this ADHD rat model.

## Effects of HD-tDCS application on protein expression of dopaminergic neurotransmission factors and NTFs in the ADHD rat model

To confirm the protein expression of dopaminergic neurotransmission factors and NTFs, such as BDNF, TGF-β1, and GDNF, we performed a western blot analysis. Similar to the gene expression findings, HD-tDCS and MPH treatment increased the expression of the dopaminergic transmission factors TH and VMAT2 and decreased the expression of DAT compared to the findings for the sham group. The tDCS-PFC group showed the most similar pattern, followed by the MPH group. Compared to the sham group results, significant changes were observed in the SN/VTA, including alterations in DAT expression in the tDCS-PFC group, in TH and DAT expression in the tDCS-M1 group, and in TH expression in the MPH group (*Table 3*, *Figure 2—figure supplement 1A*). When we considered the ratio between VMAT2 and DAT, significant changes were observed in the prefrontal cortex, striatum, and hippocampus in the tDCS-PFC group, in the hippocampus and SN/VTA in the tDCS-M1 group, and in the hippocampus in the MPH group (*Table 4*).

The tDCS-PFC group generally showed an increase in mature BDNF (mBDNF) expression in all regions except the hippocampus. A similar pattern was observed in the striatum and SN/VTA in the tDCS-M1 group, and in the striatum in the MPH group. Compared to the sham group, the tDCS-PFC group showed a significant increase in TGF-β1 in the striatum, in GDNF in the hippocampus, and in mBDNF in the SN/VTA. The tDCS-M1 group showed significant changes in mBDNF in the SN/VTA, while in the MPH group, TGF-β1 expression was changed in the striatum and GDNF expression in the hippocampus (*Table 5*, *Figure 2—figure supplement 1B*). These results suggest that HD-tDCS application over the prefrontal cortex regulates protein expression of dopaminergic neurotransmission factors and NTFs, in particular of DAT and BDNF, in this ADHD rat model.

## Effects of HD-tDCS application on free TH and mBDNF in the ADHD rat model and its control strain

To confirm free TH and mBDNF content induced by HD-tDCS, we performed an ELISA analysis in both the ADHD rat model and its WKY control strain, 2 days after the last HD-tDCS application. TH and mBDNF showed lower levels in the SHR groups compared to all WKY groups. In the ADHD rat model, free TH levels were significantly increased in the SHR-tDCS-PFC group, compared to the SHR-sham group, in the PFC (*Figure 2A*, prefrontal cortex: $F_{(5,18)}$=9.262, p<0.001). The content of mBDNF was significantly induced in the SHR-tDCS-PFC group, compared to the SHR-sham group, in the PFC and the hippocampus (*Figure 2B*, prefrontal cortex: $F_{(5,18)}$=15.327, p<0.001, hippocampus: $F_{(5,18)}$=14.924, p<0.001). However, the control strain, the WKY-tDCS-PFC group, showed no differences in free TH and mBDNF levels compared to any of the WKY and WKY-sham groups. To check for possible stress induced by HD-tDCS, we also analyzed corticosterone plasma levels. The SHR-tDCS-PFC and WKY-tDCS-PFC groups showed no changes in corticosterone plasma levels compared to the naive and the sham group (*Figure 2—figure supplement 2*, $F_{(5,18)}$=9.262, p<0.001).

To confirm that these changes were continuous, and also to compare the effect of HD-tDCS with a positive control in our ADHD rat model, we performed an additional ELISA analysis in the tDCS-PFC and MPH groups and compared the results to those obtained in the sham group at 8 days after the last HD-tDCS application. The content of TH generally increased after HD-tDCS-PFC and MPH treatment, and a significant increase was observed in the prefrontal cortex, compared to the sham results (*Figure 2C*, prefrontal cortex: $F_{(2,9)}$=17.628, p<0.001, striatum: $F_{(2,9)}$=4.036, p=0.056). The mBDNF content also showed a similar pattern following HD-tDCS-PFC and MPH treatment, and significant increases were observed in the prefrontal cortex, striatum, and SN/VTA in the tDCS-PFC group, compared to the sham results (*Figure 2D*, prefrontal cortex: $F_{(2,9)}$=12.181, p=0.008, striatum: $F_{(2,9)}$=17.204, p<0.001, hippocampus: $F_{(2,9)}$=5.964, p=0.037, SN/VTA: $F_{(2,9)}$=5.261, p=0.031). These results suggest that HD-tDCS application over the prefrontal cortex induces an increase of free TH

Table 2. Effect of HD-tDCS on gene expressions of NTFs in the prefrontal cortex, striatum, hippocampus, and SN/VTA (n = 4–5).

| Fold change (of WKY) | SHR | Sham | tDCS-PFC | tDCS-M1 | MPH | Sham vs. tDCS-PFC | Sham vs. tDCS-M1 | Sham vs. MPH | PFC vs. M1 |
|---|---|---|---|---|---|---|---|---|---|
| **Prefrontal cortex** | | | | | | | | | |
| BDNF | 1.15 ± 0.31 | 0.87 ± 0.11 | 1.23 ± 0.48 | 1.65 ± 0.39 | 0.89 ± 0.38 | p=0.146 | **p=0.003** | p=0.929 | p=0.162 |
| | | | | | | | **F(1,8)=18.279** | F(1,8)=0.00857 | F(1,8)=2.373 |
| | | | | | | F(1,8)=2.595 | | | |
| TFG-ß1 | 1.37 ± 0.35 | 1.07 ± 0.12 | 1.82 ± 0.75 | 0.88 ± 0.33 | 1.16 ± 0.27 | p=0.056 | p=0.260 | p=0.507 | **p=0.033** |
| | | | | | | | F(1,8)=1.467 | F(1,8)=0.483 | **F(1,8)=6.576** |
| GDNF | 1.13 ± 0.37 | 1.05 ± 0.23 | 0.68 ± 0.17 | 0.86 ± 0.25 | 0.80 ± 0.18 | **p=0.022** | p=0.262 | p=0.095 | **p=0.022** |
| | | | | | | **F(1,8)=8.090** | F(1,8)=1.456 | F(1,8)=3.586 | **F(1,8)=8.090** |
| NGF | 1.22 ± 0.31 | 1.51 ± 0.12 | 1.49 ± 0.30 | 1.19 ± 0.37 | 1.44 ± 0.43 | p=0.904 | p=0.104 | p=0.722 | p=0.197 |
| | | | | | | F(1,8)=0.0154 | F(1,8)=3.360 | F(1,8)=0.136 | F(1,8)=1.980 |
| NT3 | 0.69 ± 0.24 | 0.64 ± 0.12 | 0.40 ± 0.08 | 0.59 ± 0.12 | 0.51 ± 0.11 | **p=0.006** | p=0.586 | p=0.119 | **p=0.017** |
| | | | | | | **F(1,8)=13.944** | F(1,8)=0.321 | F(1,8)=3.042 | **F(1,8)=8.977** |
| **Striatum** | | | | | | | | | |
| BDNF | 1.65 ± 0.67 | 1.65 ± 0.57 | 2.69 ± 1.07 | 1.75 ± 1.02 | 0.76 ± 0.58 | p=0.090 | p=0.852 | **p=0.041** | p=0.192 |
| | | | | | | F(1,8)=3.707 | F(1,8)=0.0372 | **F(1,8)=5.884** | F(1,8)=2.035 |
| TFG-ß1 | 0.69 ± 0.42 | 0.95 ± 0.33 | 2.61 ± 0.72 | 0.94 ± 0.04 | 1.02 ± 0.21 | **p=0.002** | p=1.000 | p=0.710 | **p=0.008** |
| | | | | | | **F(1,8)=21.618** | | F(1,8)=0.148 | |
| GDNF | 0.87 ± 0.17 | 1.19 ± 0.34 | 1.66 ± 0.58 | 1.17 ± 0.31 | 1.64 ± 0.25 | p=1.2368 | p=0.927 | **p=0.046** | p=0.132 |
| | | | | | | F(1,8)=2.475 | F(1,8)=0.00895 | **F(1,8)=5.585** | F(1,8)=2.805 |
| NGF | 1.27 ± 0.68 | 1.41 ± 0.45 | 1.67 ± 0.82 | 1.19 ± 0.35 | 1.35 ± 0.34 | p=0.550 | p=0.430 | p=0.835 | p=0.270 |
| | | | | | | F(1,8)=0.390 | F(1,8)=0.691 | F(1,8)=0.0464 | F(1,8)=1.404 |
| NT3 | 0.85 ± 0.35 | 0.60 ± 0.09 | 0.25 ± 0.12 | 0.97 ± 0.32 | 0.61 ± 0.29 | **p=<0.001** | **p=0.035** | p=0.943 | **p=0.001** |
| | | | | | | **F(1,8)=27.650** | **F(1,8)=6.386** | F(1,8)=0.00554 | **F(1,8)=22.822** |
| **Hippocampus** | | | | | | | | | |
| BDNF | 1.96 ± 0.26 | 1.89 ± 0.58 | 4.27 ± 1.67 | 3.27 ± 0.70 | 2.31 ± 0.27 | **p=0.008** | **p=0.032** | p=0.183 | p=0.151 |
| | | | | | | | F(1,8)=2.122 | | |
| TFG-ß1 | 0.98 ± 0.31 | 0.74 ± 0.27 | 1.14 ± 0.31 | 1.44 ± 0.33 | 0.87 ± 0.07 | p=0.062 | **p=0.006** | p=0.690 | p=0.095 |
| | | | | | | F(1,8)=4.704 | **F(1,8)=13.400** | | |
| GDNF | 1.03 ± 0.23 | 0.72 ± 0.11 | 1.67 ± 0.59 | 0.82 ± 0.17 | 0.84 ± 0.10 | **p=0.029** | p=0.384 | p=0.173 | **p=0.029** |
| | | | | | | | F(1,6)=0.882 | F(1,6)=2.387 | |
| NGF | 1.16 ± 0.30 | 0.91 ± 0.31 | 2.85 ± 0.92 | 2.46 ± 0.23 | 1.59 ± 0.20 | **p=0.007** | **P=<0.001** | **p=0.011** | p=0.448 |
| | | | | | | **F(1,6)=15.973** | **F(1,6)=63.722** | **F(1,6)=13.366** | F(1,6)=0.660 |
| NT3 | 1.17 ± 0.15 | 1.09 ± 0.27 | 1.77 ± 0.75 | 1.74 ± 0.27 | 1.42 ± 0.15 | p=0.093 | **p=0.005** | **p=0.041** | p=0.935 |
| | | | | | | F(1,8)=3.637 | **F(1,8)=14.400** | **F(1,8)=5.931** | F(1,8)=0.00709 |
| **SN/VTA** | | | | | | | | | |
| BDNF | 1.02 ± 0.18 | 1.28 ± 0.13 | 1.74 ± 0.42 | 1.69 ± 0.33 | 1.42 ± 0.54 | p=0.114 | p=0.062 | p=0.641 | p=0.846 |
| | | | | | | | F(1,6)=5.217 | F(1,6)=0.241 | F(1,6)=0.0412 |
| TFG-ß1 | 0.49 ± 0.14 | 0.56 ± 0.09 | 0.86 ± 0.12 | 0.63 ± 0.12 | 0.75 ± 0.19 | **p=0.008** | p=0.373 | p=0.119 | **p=0.042** |
| | | | | | | **F(1,6)=14.868** | F(1,6)=0.928 | F(1,6)=3.303 | **F(1,6)=6.629** |
| GDNF | 0.52 ± 0.04 | 0.57 ± 0.04 | 1.00 ± 0.26 | 0.51 ± 0.05 | 0.96 ± 0.16 | p=0.156 | p=0.428 | p=0.061 | p=0.118 |
| | | | | | | F(1,6)=2.635 | F(1,6)=0.723 | F(1,6)=5.299 | F(1,6)=3.331 |
| NGF | 1.24 ± 0.14 | 0.93 ± 0.07 | 0.62 ± 0.41 | 0.98 ± 0.35 | 1.24 ± 0.98 | p=0.188 | p=0.764 | p=0.486 | p=0.200 |
| | | | | | | F(1,6)=2.209 | F(1,6)=0.0991 | | |
| NT3 | 1.13 ± 0.35 | 1.21 ± 0.07 | 0.79 ± 0.48 | 1.55 ± 0.96 | 1.45 ± 0.60 | p=0.343 | p=0.486 | p=1.000 | p=0.486 |

Expression of the *BDNF*, *TGF-β1*, *GDNF*, *NGF*, and *NT3* genes, expressed as fold changes of WKY. Data represent the mean ± SEM. Data were analyzed using ANOVA for repeated measures, followed by Tukey's tests for multiple comparisons. p<0.05 was considered statistically significant; significant results are highlighted in bold.

and mBDNF content in this ADHD rat model, but not in the WKY control strain, and that these content changes were continuous.

## Effects of HD-tDCS application on TH- and DAT-positive cells in the ADHD rat model

We first verified whether HD-tDCS induces a neuroinflammatory response in the brain tissue. The observed changes in activated microglia stained with ionized calcium-binding adaptor molecule 1 (Iba1) indicates that the charge density of anodal HD-tDCS application cannot trigger an

**Table 3.** Effect of HD-tDCS on protein expression of dopaminergic neurotransmission factors in the prefrontal cortex, striatum, hippocampus, and SN/VTA (n = 5).

| % of WKY | SHR | Sham | tDCS-PFC | tDCS-M1 | MPH | Sham vs. tDCS-PFC | Sham vs. tDCS-M1 | Sham vs. MPH | PFC vs. M1 |
|---|---|---|---|---|---|---|---|---|---|
| **Prefrontal cortex** | | | | | | | | | |
| TH | 53.52 ± 31.75 | 82.28 ± 43.03 | 100.94 ± 26.54 | 98.30 ± 43.88 | 65.79 ± 18.48 | p=0.43 | p=0.58 | p=0.45 | p=0.91 |
| | | | | | | F(1,8)=0.68 | F(1,8)=0.34 | F(1,8)=0.62 | F(1,8)=0.01 |
| DAT | 89.41 ± 52.58 | 108.69 ± 31.93 | 89.63 ± 39.41 | 83.32 ± 32.20 | 86.40 ± 27.72 | p=0.43 | p=0.25 | p=0.27 | p=0.79 |
| | | | | | | F(1,8)=0.70 | F(1,8)=1.56 | F(1,8)=1.39 | F(1,8)=0.08 |
| VMAT2 | 71.61 ± 48.26 | 57.09 ± 24.08 | 76.33 ± 20.62 | 71.56 ± 40.23 | 70.45 ± 29.33 | p=0.21 | p=0.51 | p=0.45 | p=0.82 |
| | | | | | | F(1,8)=1.84 | F(1,8)=0.48 | F(1,8)=0.62 | F(1,8)=0.06 |
| **Striatum** | | | | | | | | | |
| TH | 103.70 ± 15.14 | 95.96 ± 11.46 | 101.56 ± 22.61 | 91.96 ± 11.52 | 103.76 ± 21.11 | p=0.63 | p=0.55 | p=0.49 | p=0.42 |
| | | | | | | F(1,8)=0.24 | | F(1,8)=0.53 | F(1,8)=0.72 |
| DAT | 117.76 ± 37.47 | 101.44 ± 34.52 | 120.34 ± 71.05 | 101.35 ± 47.65 | 107.57 ± 34.41 | p=0.61 | p=1.0 | p=0.79 | p=0.63 |
| | | | | | | F(1,8)=0.29 | | F(1,8)=0.08 | F(1,8)=0.25 |
| VMAT2 | 90.46 ± 57.19 | 66.67 ± 38.56 | 71.25 ± 35.22 | 85.87 ± 56.83 | 73.78 ± 25.00 | p=0.85 | p=0.55 | p=0.74 | p=0.64 |
| | | | | | | F(1,8)=0.04 | F(1,8)=0.39 | F(1,8)=0.12 | F(1,8)=0.24 |
| **Hippocampus** | | | | | | | | | |
| TH | 79.55 ± 40.22 | 111.39 ± 31.31 | 78.36 ± 45.70 | 96.29 ± 52.81 | 78.48 ± 46.69 | p=0.22 | p=0.60 | p=0.23 | p=0.58 |
| | | | | | | F(1,8)=1.78 | F(1,8)=0.30 | F(1,8)=1.71 | F(1,8)=0.33 |
| DAT | 64.33 ± 35.74 | 159.83 ± 87.35 | 87.51 ± 56.22 | 152.10 ± 68.80 | 77.47 ± 28.67 | p=0.16 | p=0.88 | p=0.08 | p=0.14 |
| | | | | | | F(1,8)=2.42 | F(1,8)=0.024 | F(1,8)=4.01 | F(1,8)=2.64 |
| VMAT2 | 81.40 ± 30.60 | 88.84 ± 26.80 | 113.49 ± 33.17 | 91.26 ± 17.67 | 117.83 ± 22.95 | p=0.23 | p=0.87 | p=0.10 | p=0.22 |
| | | | | | | F(1,8)=1.67 | F(1,8)=0.028 | F(1,8)=3.38 | F(1,8)=1.75 |
| **SN/VTA** | | | | | | | | | |
| TH | 84.35 ± 48.89 | 68.41 ± 30.38 | 94.69 ± 34.28 | 120.88 ± 31.89 | 142.75 ± 58.04 | p=0.24 | **p=0.03** | **p=0.04** | p=0.25 |
| | | | | | | F(1,8)=1.65 | **F(1,8)=7.10** | **F(1,8)=6.44** | F(1,8)=1.56 |
| DAT | 56.52 ± 27.67 | 96.48 ± 21.33 | 40.37 ± 29.72 | 56.96 ± 27.14 | 72.24 ± 47.04 | **p=0.01** | **p=0.03** | p=0.33 | p=0.38 |
| | | | | | | **F(1,8)=11.76** | **F(1,8)=6.56** | F(1,8)=1.10 | F(1,8)=0.85 |
| VMAT2 | 56.87 ± 33.03 | 74.15 ± 31.39 | 79.95 ± 25.70 | 53.48 ± 17.09 | 57.24 ± 28.47 | p=0.76 | p=0.23 | p=0.40 | p=0.09 |
| | | | | | | F(1,8)=0.10 | F(1,8)=1.67 | F(1,8)=0.80 | F(1,8)=3.68 |

Expression of the TH, DAT, and VMAT2 proteins, expressed as percentages of WKY. Data are presented as the mean ± SEM. Data were analyzed using ANOVA for repeated measures, followed by Tukey's tests for multiple comparisons. p<0.05 was considered statistically significant; significant results are highlighted in bold.

**Table 4.** Effect of HD-tDCS on the ratio between VMAT2 and DAT protein in the prefrontal cortex, striatum, hippocampus, and SN/VTA (n = 4).

| VMAT2/DAT ratio | WKY | SHR | Sham | tDCS-PFC | tDCS-M1 | MPH | Sham vs. tDCS-PFC | Sham vs. tDCS-M1 | Sham vs. MPH | PFC vs. M1 |
|---|---|---|---|---|---|---|---|---|---|---|
| Prefrontal cortex | 2.80 ± 0.98 | 2.05 ± 1.24 | 1.35 ± 0.22 | 2.61 ± 0.06 | 2.61 ± 0.41 | 1.66 ± 0.23 | **p=0.029** | p=0.939 | p=0.108 | **p<0.001** |
| | | | | | | | | | F(1,6)=3.55 | **F(1,6)=35.56** |
| Striatum | 2.71 ± 0.81 | 1.32 ± 1.13 | 1.16 ± 0.71 | 2.96 ± 1.27 | 1.94 ± 0.96 | 1.53 ± 0.15 | **p=0.048** | p=0.240 | p=0.358 | p=0.200 |
| | | | | | | | **F(1,6)=6.11** | F(1,6)=1.70 | F(1,6)=0.99 | |
| Hippocampus | 2.44 ± 1.25 | 1.31 ± 0.32 | 0.60 ± 0.10 | 1.68 ± 0.68 | 1.14 ± 0.33 | 1.86 ± 0.58 | **p=0.029** | **p=0.029** | **p=0.029** | p=0.207 |
| | | | | | | | | | | F(1,6)=2.00 |
| SN/VTA | 0.91 ± 0.16 | 0.19 ± 0.06 | 0.42 ± 0.09 | 1.12 ± 0.69 | 1.14 ± 0.58 | 0.60 ± 0.31 | p=0.114 | **p=0.049** | p=0.303 | p=0.114 |
| | | | | | | | | **F(1,6)=6.03** | F(1,6)=1.27 | |

Data are presented as the mean ± SEM. Data were analyzed using ANOVA for repeated measures, followed by Tukey's tests for multiple comparisons. p<0.05 was considered statistically significant; significant results are highlighted in bold.

inflammatory response but rather reduces it in the PFC (*Figure 3—figure supplement 1*, prefrontal cortex: $F_{(2,12)}$=13.008, p<0.001, primary motor cortex: $F_{(2,12)}$=4.541, p=0.034). Then, we performed immunofluorescence to reveal the distribution of TH- and DAT-positive cells in the tDCS-PFC and MPH groups, and compared the results to the distribution observed in the sham animals. The mean integral optical density (IOD) of TH-positive cells was significantly increased in the medial prefrontal cortex and striatum in the tDCS-PFC and MPH groups, while IOD in the nucleus accumbens (NAc) core of the MPH group was similar to that observed in the sham animals. The IOD of DAT was also significantly reduced in the medial prefrontal cortex, dorsal striatum, and SN/VTA in the tDCS-PFC group. The IOD of TH/DAT double-positive cells showed a marked decrease in the NAc core and SN/VTA in the tDCS-PFC group and in the NAc core in the MPH group (*Figure 3*, dorsal striatum: TH; $F_{(2,12)}$=7.667, p=0.007, NAc core: TH; $F_{(2,12)}$=21.182, p<0.001, hippocampus: TH; $F_{(2,12)}$=8.988, p=0.004, SN: DAT; $F_{(2,12)}$=11.406, p=0.002, TH/DAT; $F_{(2,12)}$=6.015, p=0.016, VTA: DAT; $F_{(2,12)}$=7.874, p=0.007, TH/DAT; $F_{(2,12)}$=9.477, p=0.003). When we further divided the VTA into medial and lateral regions, sham animals demonstrated a higher IOD of TH/DAT double-positive cells in the lateral region. However, these values were significantly reduced in the tDCS-PFC group, compared to the values observed in sham animals (*Figure 3—figure supplement 2*, sham: $F_{(4,3)}$=10.116, p=0.001, vlVTA: $F_{(2, 12)}$=4.302, p=0.039). Together, these results suggest that HD-tDCS over the prefrontal cortex may enhance dopaminergic neurotransmission by down-regulation of DAT, especially in the prefrontal cortex, striatum, and SN/VTA.

## Effects of HD-tDCS application on BDNF- and its activated receptor-positive cells in the ADHD rat model

Lastly, we performed immunofluorescence for mBDNF and its activated phospho-tropomyosin receptor kinase B (pTrkB). In the tDCS-PFC group, the numbers of mBDNF-, pTrkB-, and mBDNF/pTrkB double-positive cells were significantly increased in all target regions except the SN. A similar pattern was also observed in the medial prefrontal cortex and the NAc core in the MPH group, with more mBDNF/pTrkB double-positive cells in the dorsal striatum (*Figure 4*, medial prefrontal cortex: mBDNF; $F_{(2,12)}$=17.827, p<0.001, pTrkB; $F_{(2,12)}$=55.806, p<0.001, mBDNF/pTrkB; $F_{(2, 12)}$=15.605, p<0.001, dorsal striatum: mBDNF; $F_{(2, 12)}$=15.951, p<0.001, pTrkB; $F_{(2, 12)}$=17.250, p<0.001, mBDNF/pTrkB; $F_{(2, 12)}$=20.562, p<0.001, NAc core: mBDNF; $F_{(2, 12)}$=28.260, p<0.001, pTrkB; $F_{(2, 12)}$=22.552, p<0.001, mBDNF/pTrkB; $F_{(2, 12)}$=23.163, p<0.001, hippocampus: mBDNF/pTrkB; $F_{(2, 12)}$=8.227, p=0.006, VTA: mBDNF; $F_{(2, 12)}$=27.217, p<0.001, pTrkB; $F_{(2, 12)}$=12.689, p=0.001, mBDNF/pTrkB; $F_{(2,12)}$=21.925, p<0.001). Considering that hippocampal neurogenesis is closely related to BDNF expression, we examined bromodeoxyuridine (BrdU)-positive and BrdU/neuronal nuclei (NeuN) double-positive cells in the dentate gyrus of the hippocampus. BrdU-positive cells increased following HD-tDCS and MPH treatment, and a significant increase was observed in the number of BrdU/NeuN double-positive cells in the tDCS-PFC group compared to the sham group (*Figure 5*). These results suggest that HD-tDCS application over prefrontal cortex significantly

**Table 5.** Effect of HD-tDCS on protein expression of NTFs in the prefrontal cortex, striatum, hippocampus, and SN/VTA (n = 5).

| % of WKY | SHR | Sham | tDCS-PFC | tDCS-M1 | MPH | Sham vs. tDCS-PFC | Sham vs. tDCS-M1 | Sham vs. MPH | PFC vs. M1 |
|---|---|---|---|---|---|---|---|---|---|
| Prefrontal cortex | | | | | | | | | |
| mBDNF | 46.16 ± 13.97 | 48.99 ± 26.00 | 82.54 ± 20.37 | 72.62 ± 19.36 | 51.81 ± 14.08 | p=0.053 | p=0.14 | p=0.84 | p=0.45 |
| | | | | | | F(1,8)=5.16 | F(1,8)=2.66 | F(1,8)=0.05 | F(1,8)=0.62 |
| TFG-ß1 | 36.61 ± 15.36 | 68.19 ± 34.34 | 76.41 ± 31.12 | 63.55 ± 19.22 | 56.20 ± 31.88 | p=0.70 | p=0.80 | p=0.58 | p=0.45 |
| | | | | | | F(1,8)=0.16 | F(1,8)=0.07 | F(1,8)=0.33 | F(1,8)=0.62 |
| Striatum | | | | | | | | | |
| mBDNF | 70.62 ± 13.01 | 58.01 ± 20.04 | 75.05 ± 15.97 | 81.41 ± 18.36 | 94.96 ± 38.41 | p=0.18 | p=0.09 | p=0.09 | p=0.58 |
| | | | | | | F(1,8)=2.21 | F(1,8)=3.71 | F(1,8)=3.64 | F(1,8)=0.34 |
| TFG-ß1 | 75.83 ± 31.20 | 61.40 ± 18.32 | 96.49 ± 16.73 | 81.31 ± 36.90 | 95.82 ± 19.27 | **p=0.01** | p=0.31 | **p=0.02** | p=0.43 |
| | | | | | | **F(1,8)=10.00** | F(1,8)=1.17 | **F(1,8)=8.38** | F(1,8)=0.70 |
| Hippocampus | | | | | | | | | |
| mBDNF | 48.39 ± 20.57 | 78.16 ± 18.68 | 76.51 ± 19.73 | 82.80 ± 9.07 | 72.31 ± 24.47 | p=0.90 | p=0.63 | p=0.68 | p=0.54 |
| | | | | | | F(1,8)=0.02 | F(1,8)=0.25 | F(1,8)=0.18 | F(1,8)=0.42 |
| TFG-ß1 | 42.67 ± 18.24 | 70.02 ± 27.53 | 104.94 ± 67.08 | 53.83 ± 31.95 | 94.76 ± 64.33 | p=0.31 | p=0.42 | p=0.45 | p=0.16 |
| | | | | | | F(1,8)=1.16 | F(1,8)=0.74 | F(1,8)=0.63 | F(1,8)=2.37 |
| GDNF | 129.07 ± 53.16 | 181.27 ± 22.52 | 322.41 ± 109.41 | 182.26 ± 43.25 | 235.44 ± 42.26 | **p=0.02** | p=0.97 | **p=0.04** | **p=0.03** |
| | | | | | | **F(1,8)=7.98** | | **F(1,8)=6.40** | **F(1,8)=7.10** |
| SN/VTA | | | | | | | | | |
| mBDNF | 68.10 ± 17.65 | 76.63 ± 8.76 | 102.87 ± 15.04 | 103.20 ± 8.12 | 100.30 ± 30.01 | **p=0.01** | **p=0.001** | p=0.13 | p=0.97 |
| | | | | | | **F(1,8)=11.36** | **F(1,8)=24.74** | F(1,8)=2.87 | |
| TFG-ß1 | 72.73 ± 26.35 | 77.68 ± 36.44 | 100.19 ± 45.32 | 111.19 ± 16.46 | 77.67 ± 49.81 | p=0.41 | p=0.10 | p=1.00 | p=0.624 |
| | | | | | | F(1,8)=0.75 | F(1,8)=3.51 | | F(1,8)=0.26 |

Expression of the mBDNF, TFG-ß1, and GDNF proteins, expressed as percentages of WKY. Data are presented as the mean ± SEM. Data were analyzed using ANOVA for repeated measures, followed by Tukey's tests for multiple comparisons. p<0.05 was considered statistically significant; significant results are highlighted in bold.

induced activation of BDNF-TrkB signaling in most examined brain regions. Moreover, HD-tDCS application activated hippocampal neurogenesis.

## Discussion

We investigated the therapeutic effects of multi-day sessions of HD-tDCS on recovery from cognitive symptoms observed in a rat model of ADHD. The main findings of the present study are the following: (1) application of anodal HD-tDCS over the prefrontal cortex was associated with beneficial effects on cognition; (2) application of anodal HD-tDCS increased dopaminergic signaling factors TH and VMAT2, and significantly decreased DAT; (3) application of anodal HD-tDCS led to an increase in several NTFs, such as BDNF, TGF-β1, and GDNF. In particular, BDNF showed the most significant increase and was associated with hippocampal neurogenesis. Taken together, our findings suggest that anodal HD-tDCS over the prefrontal cortex regulates dopaminergic neurotransmission and NTFs and that it therefore provides a potential therapeutic tool for cognitive dysfunction in patients with ADHD.

ADHD is a heterogeneous neuropsychiatric disorder associated with impaired cognitive function, including problems with behavioral flexibility (*Cui et al., 2018*; *Faraone et al., 2015*; *Tsai, 2017*). Recent studies suggest that non-invasive brain stimulation with tDCS may have therapeutic potential for improving cognitive impairments in patients with ADHD (*Coffman et al., 2014*; *Leffa et al., 2018*; *Leffa et al., 2016*; *Wu et al., 2017*). The major concerns associated with tDCS treatment are related to its safeness as well as to cortical excitability. Two generalized safety parameters for

dosing, electrode current, and charge density have been suggested (*Jackson et al., 2017*; *Liebetanz et al., 2009*), and our electrode montage provides an additional safety margin (*Jackson et al., 2017*; *Liebetanz et al., 2009*). More than 80% of the applied current is lost in the rat; enough current flowing from the electrode is, however, essential to elicit cortical electric fields (*Vöröslakos et al., 2018*). Our computational models predicted that values for both electric field intensity and current density would peak at the prefrontal or primary motor cortex according to the electrode position. Electric fields as low as ~1 V/m can affect the timing of action potentials; however, higher intensities (~2 V/m) are required to measurably affect local field potential and the associated brain networks (*Liu et al., 2018*; *Vöröslakos et al., 2018*). Our predicted peak electric field intensity was slightly lower than the values in previous rodent experiments, which reported an average intracranial field intensity of 6.8 ± 3.8 V/m (*Liu et al., 2018*). It also predicted the absence of brain lesions, in line with animal models indicating that brain injury occurs at values of 6.3–13 A/m$^2$ (*Antal et al., 2017*; *Bikson et al., 2016*).

We first investigated whether HD-tDCS improves cognitive function and flexibility in our rat model of ADHD. We found that HD-tDCS application over the prefrontal cortex exerted a greater beneficial effect than HD-tDCS over the primary motor cortex. HD-tDCS over the prefrontal cortex reduced hyperactivity and alleviated cognitive dysfunctions in spatial learning and working memory, behavioral flexibility, and discrimination and exploratory behavior. These effects were similar to those in the positive control group receiving MPH treatment, with even greater benefits with respect to discrimination ability.

According to the 'compartment concept' of dopamine action, extracellular space is considered a site of potential action; thus inhibition of intracellular uptake (or enhancement of vesicular transporters) has been regarded as a target for therapeutic trials (*Uhl, 1998*). In the dopaminergic neurotransmission system, DAT modulates levels of extracellular and intracellular dopamine stores by controlling reuptake from the extracellular space (*Jones et al., 1998*; *Leo et al., 2018*). VMAT2 is a transporter protein expressed on vesicular membranes for monoamines such as dopamine and modulates dopamine release dynamics (*Sandoval et al., 2002*; *Uhl, 1998*). DAT and VMAT2 are main regulators of dopamine homeostasis and signaling dynamics between extra- and intracellular compartments in the brain (*Guillot and Miller, 2009*).

We analyzed gene and protein expression for dopaminergic neurotransmission factors in the core regions of the mesocorticolimbic dopamine pathway. HD-tDCS application induced increases in TH and VMAT2 gene and protein expression and a decrease in DAT expression, a pattern that was most pronounced following HD-tDCS application over the prefrontal cortex. DAT is considered the main target for stimulant drugs used to treat ADHD, such as methylphenidate, a DAT antagonist that works by elevating extracellular dopamine levels (*Bonvicini et al., 2016*). The ratio between DAT and VMAT2, key regulators of dopaminergic neurotransmission, is a useful molecular marker for dopamine homeostasis; that is, elevated levels of VMAT2 relative to DAT protein translate to higher extracellular levels and, presumably, decreased cytosolic levels of dopamine (*Guillot and Miller, 2009*; *Lohr et al., 2014*; *Uhl, 1998*). HD-tDCS application over the prefrontal cortex showed significant changes in VMAT2 relative to DAT in the prefrontal cortex, striatum, and hippocampus, suggesting possible higher extracellular levels of dopamine in these regions.

NTFs related to neuronal development and plasticity are believed to be involved in the clinical manifestation of ADHD (*Banaschewski et al., 2010*; *Galvez-Contreras et al., 2017*; *Tsai, 2017*). One strong candidate that may play a role in the pathogenesis of ADHD is BDNF (*Kent et al., 2005*). *Bdnf* knockout mice exhibit fundamental behavioral characteristics of ADHD (*Kernie et al., 2000*), and drugs that can activate BDNF signaling have shown therapeutic potential for ADHD (*Tsai, 2017*). HD-tDCS application induced the upregulation of several NTF genes, including a significant upregulation of *BDNF*, *TGF-β1*, and *GDNF*. In particular, HD-tDCS application over the prefrontal cortex generally induced an increase in BDNF gene and protein expression in all brain regions examined except the hippocampus. Significant induction of BDNF protein was also observed in the SN/VTA of HD-tDCS-treated mice. ADHD rats exhibited decreased BDNF expression in the hippocampus, and impaired spatial learning ability (*Jeong et al., 2014*). BDNF secretion is essential for learning and memory in tDCS application (*Fritsch et al., 2010*; *Podda et al., 2016*). Similarly, our study found that BDNF expression was more markedly increased by HD-tDCS application compared to changes in the other examined NTFs. We confirmed TH and BDNF protein content by ELISA and found that HD-tDCS application induced a significant and continuous increase, not

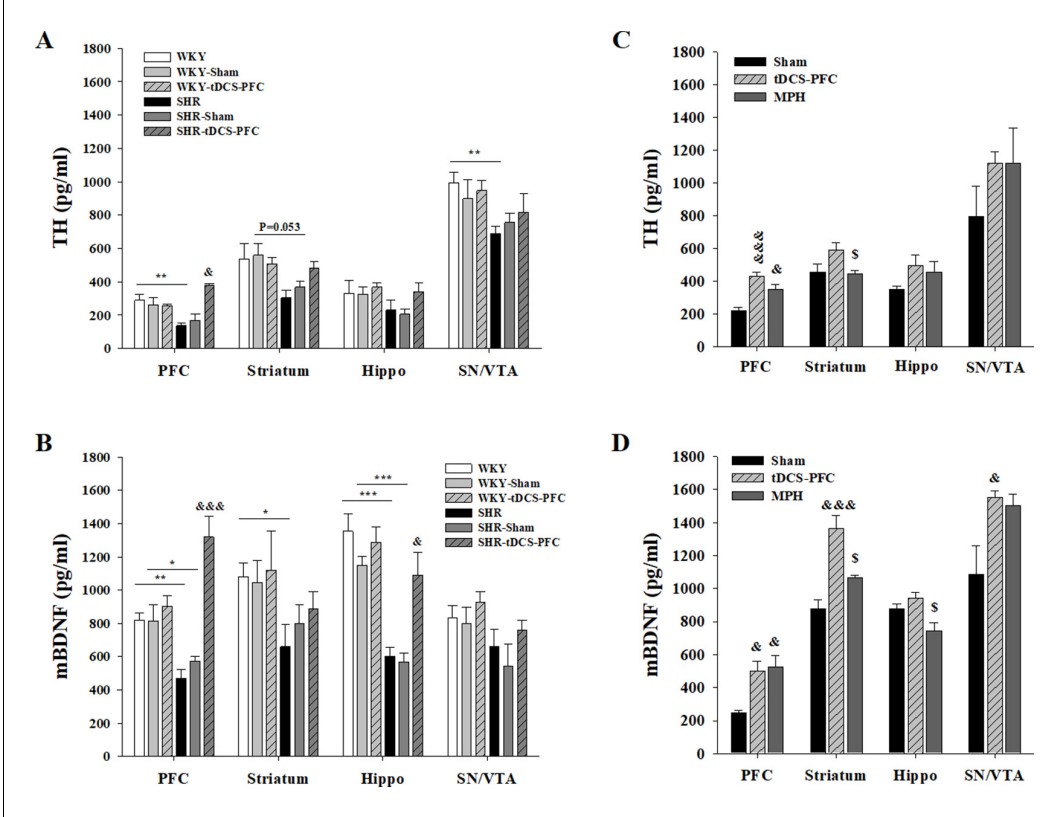

**Figure 2.** Effect of HD-tDCS application over the prefrontal cortex on free TH and mBDNF levels. (**A, B**) Free TH and mBDNF levels measured by ELISA at 2 days after the last HD-tDCS application in our ADHD rat model and its WKY control strain (n = 4). In the ADHD rat model, TH and mBDNF levels were significantly increased in the prefrontal cortex in the SHR-tDCS-PFC group compared to the sham group. mBDNF levels were also significantly increased in the hippocampus in the SHR-tDCS-PFC group. (**C, D**) Free TH and mBDNF levels measured by ELISA at 8 days after the last HD-tDCS application in our rat model of ADHD (n = 4). Levels of TH were significantly increased in the prefrontal cortex in the tDCS-PFC and MPH-treated groups compared to the sham group. mBDNF levels were markedly increased in the prefrontal cortex, striatum, and SN/VTA in the tDCS-PFC group compared to the sham group. Data are presented as the mean ± SEM. [&]$p<0.05$ and [&&&]$p<0.001$ vs. sham, [$]$p<0.05$ vs. tDCS-PFC.

The online version of this article includes the following source data and figure supplement(s) for figure 2:

**Source data 1.** Source files for quantification of ELISA analysis.

**Figure supplement 1.** Effect of HD-tDCS on protein expression of dopaminergic neurotransmission factors and NTFs in a rat model of ADHD.

**Figure supplement 2.** Effect of HD-tDCS application over the prefrontal cortex on corticosterone levels in the plasma of our ADHD rat model and its control strain at 2 days after the last HD-tDCS application.

observed in the WKY control stain, in TH levels in the prefrontal cortex and in mBDNF levels in the prefrontal cortex, striatum, and SN/VTA.

Finally, we performed immunofluorescence to observe the distribution of TH/DAT-positive cells, the main target for therapeutic drugs for ADHD (*Bonvicini et al., 2016*; *Uhl, 1998*), and of BDNF/ pTrkB-positive cells, which have shown therapeutic potential in learning and memory in ADHD (*Fritsch et al., 2010*; *Tsai, 2017*). Similar to the results on gene and protein expressions, HD-tDCS application over the prefrontal cortex induced a significant increase in TH-positive cells in the medial prefrontal cortex and the striatum. A marked decrease in DAT-positive cells was observed in the medial prefrontal cortex, dorsal striatum, and SN/VTA. Dopaminergic innervations from the VTA mainly target the prefrontal cortex and ventral striatum (NAc), and also innervate other regions such as the hippocampus. Thus, dopamine signaling represents a crucial factor modulating executive functions including cognitive flexibility, incentive motivation, and reward processing (*Russo and*

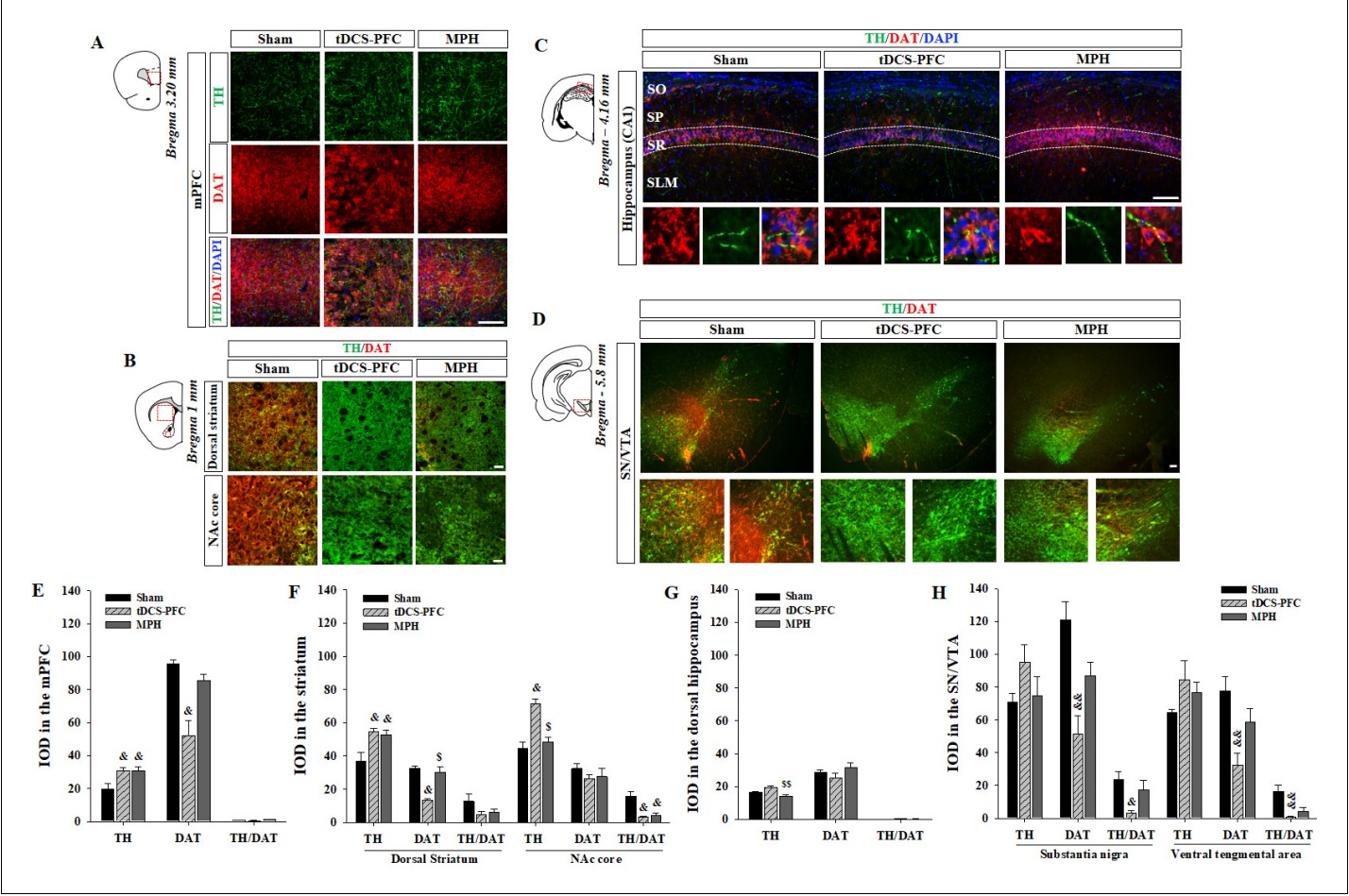

**Figure 3.** Effect of HD-tDCS application over the prefrontal cortex on TH- and DAT-positive cells in a rat model of ADHD. (A, E) Photomicrograph and histogram showing the mean IOD of TH- and DAT-positive cells in the medial prefrontal cortex, (B, F) in the striatum, (C, G) in the dorsal hippocampus, and (D, H) in the SN/VTA. The IOD of TH-positive cells was significantly increased in the medial prefrontal cortex and striatum of tDCS-PFC group compared to the sham results. The IOD of DAT-positive cells was significantly decreased in the medial prefrontal cortex, dorsal striatum, and SN/VTA in the tDCS-PFC group compared to the sham results. Data are presented as the mean ± SEM. mPFC, medial prefrontal cortex. [&]p<0.05 and [&&]p<0.01 vs. sham, [$]p<0.05 and [$$]p<0.01 vs. tDCS-PFC. Scale bar = 100 µm.

The online version of this article includes the following source data and figure supplement(s) for figure 3:

**Source data 1.** Source file for quantification of Iba1-, TH-, and DAT-positive cells.

**Figure supplement 1.** Neuroinflammatory analysis for chronic HD-tDCS application in our rat model of ADHD.

**Figure supplement 2.** Effect of HD-tDCS application over the prefrontal cortex on the expression of TH/DAT double-positive cells in the VTA in a rat model of ADHD.

*Nestler, 2013*). Midbrain dopaminergic neurons also project to the striatum and NAc topographically along the mediolateral axis; thus, lateral VTA neurons that project to the NAc core influence motor responses related to reward stimuli (*Nobili et al., 2017*). There were significantly fewer DAT-positive cells in the prefrontal cortex and striatum and TH/DAT double-positive cells in the lateral region of the VTA following HD-tDCS application, suggesting that HD-tDCS affects executive functions through the regulation of dopaminergic neurotransmission in the prefrontal cortex and the NAc.

HD-tDCS application induced the activation of BDNF/TrkB, even more than MPH treatment, in all target regions except the SN. Enhancing adult hippocampal neurogenesis improves cognitive function and is closely related to BDNF expression (*Choi et al., 2018*). Thus, we quantified neurogenesis in the dentate gyrus of the hippocampus and found a significant increase in newly formed neurons following HD-tDCS application. A comparison of the HD-tDCS group with the positive control group receiving DAT antagonist MPH revealed similar therapeutic effects. Indeed, HD-tDCS application

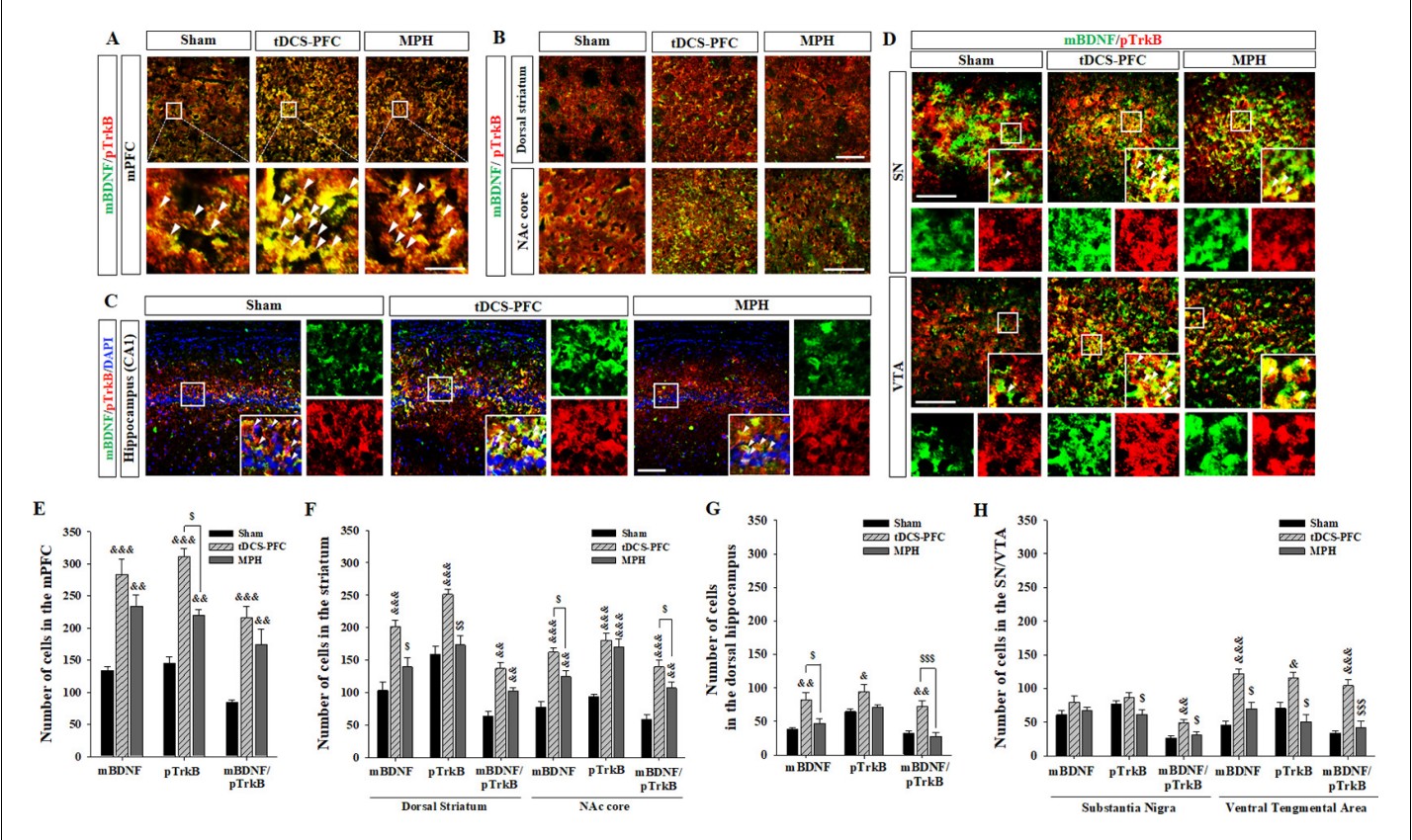

**Figure 4.** Effects of HD-tDCS application over the prefrontal cortex on mBDNF- and pTrkB-positive cells in our rat model of ADHD. (A, E) Photomicrograph and histogram showing the mean number of mBDNF- and pTrkB-positive cells in the medial prefrontal cortex, (B, F) in the striatum, (C, G) in the dorsal hippocampus, and (D, H) in the SN/VTA. The mBDNF- and pTrkB-positive cells were significantly increased in the tDCS-PFC group compared to the sham animals at all sites except the SN. Significantly more mBDNF/pTrkB double-positive cells were also detected in the SN. Data are presented as the mean ± SEM. [&]p<0.05, [&&]p<0.01 and [&&&]p<0.001 vs. sham; [$]p<0.05, [$$]p<0.01 and [$$$]p<0.001 vs. tDCS-PFC. Scale bar = 100 μm. The online version of this article includes the following source data for figure 4:

**Source data 1.** Source files for quantification of BDNF and its activated receptor.

over the prefrontal cortex had a more significant effect on BDNF expression than MPH. This suggests that long-term application of tDCS may mediate N-methyl-D-aspartate (NMDA) receptors and increase intracellular calcium levels that link to the downstream molecular cascades, leading to the secretion of activity-dependent growth factors such as BDNF (*Liebetanz et al., 2002*; *Nitsche et al., 2003*; *Wu et al., 2017*). BDNF also enhances dopamine release in the NAc through the activation of TrkB receptors on dopaminergic nerve terminals, and is thus considered a key regulator of the meso-limbic dopamine pathway linking cognitive processes (*Berton et al., 2006*; *Goggi et al., 2003*). Our results shed light on the possibility that the activation of the BDNF/TrkB pathway may be involved in dopamine release at the core sites of dopaminergic pathways.

Non-invasive brain stimulators generate neuronal activation at the primary target site and then spread to subcortical sites of the mesocorticolimbic dopamine system via mediating NMDA receptors that induce intracellular calcium cascades (*Diana et al., 2017*; *Ferenczi and Deisseroth, 2016*; *Liebetanz et al., 2002*; *Nitsche et al., 2003*; *Wu et al., 2017*). The observed connectivity-based spread from the prefrontal cortex to subcortical sites following HD-tDCS may be related to the therapeutic effects on cognitive deficits in patients with ADHD. Our results show that similar therapeutic effects in the tDCS-PFC and the tDCS-M1 groups, potentially due to connectivity-based spread from the prefrontal cortex to subcortical sites (*Liebetanz et al., 2002*; *Nitsche et al., 2003*; *Wu et al., 2017*) and direct electrical stimulation of subcortical structures (*Hadar et al., 2020*; *Thibaut et al., 2015*). However, other possible mechanisms should be considered that the stimulation of HD-tDCS

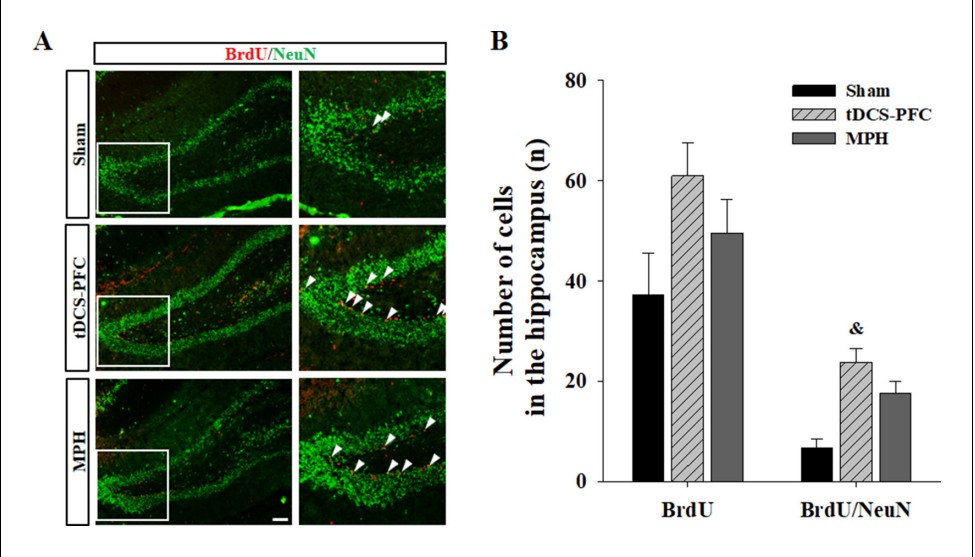

**Figure 5.** Effect of HD-tDCS application over the prefrontal cortex on hippocampal neurogenesis in our rat model of ADHD. (**A, B**) Photomicrograph and histogram showing the mean number of BrdU - and NeuN-positive cells in the dentate gyrus of the hippocampus. The number of BrdU/NeuN double-positive cells was significantly higher in the tDCS-PFC group compared to the sham animals. Data are presented as the mean ± SEM. &p<0.05 vs. sham. Scale bar = 100 µm.

The online version of this article includes the following source data for figure 5:

**Source data 1.** Source files for quantification of BrdU- and BrdU/NeuN-double-positive cells.

may partially induce cortical excitability via transcutaneous peripheral nerve stimulation, depending on the location of the active and reference electrode (*Asamoah et al., 2019*; *Tsuiki et al., 2019*).

While our study suggests that tDCS application provides a potential adjuvant therapy for ADHD, our findings have some limitations regarding their transfer to the clinical practice. The charge density and predicted current density was much higher than what is currently applied in patients, which makes it difficult to directly transfer our approach to clinical situations (*Bikson et al., 2016*; *Liebetanz et al., 2009*). However, head size needs to be considered regarding the transfer of stimulation parameters and dose regimes need to be adapted to achieve comparable conditions in animal and clinical studies. The relative difference in electric field strength between the mouse and humans is 100 times, with a typical transcranial electric stimulation intensity of 1–2 mA in humans approximately translating to 10–20 µA in mice (*Alekseichuk et al., 2019*). Compared to the intensity of <1 V/m measured in conventional human studies (*Liu et al., 2018*), rodent studies typically employ stronger intensities. The human brain also requires a higher field intensity than that used in conventional tDCS to reliably affect local field potential (*Liu et al., 2018*; *Vöröslakos et al., 2018*). Our results encourage the development of intensified tDCS protocols to induce more stable or more potent, that is, therapeutically longer-lasting, benefits of neuromodulation in psychiatric disorders (*Antal et al., 2017*; *Jackson et al., 2016*; *Liebetanz et al., 2009*).

Consequently, our results present evidence of the therapeutic potential of anodal HD-tDCS over the prefrontal cortex in the treatment of ADHD. This treatment regulated dopaminergic neurotransmission factors (especially DAT) and BDNF signaling in the core dopaminergic pathways, subsequently ameliorating cognitive dysfunction. The results of our gene and protein expression analysis, that is, TGF-β1 protein in the striatum and GDNF protein in the hippocampus indicate that NTFs such as TGF-β1 and GDNF are also activated; future research that aims to clarify these mechanisms is warranted. In summary anodal HD-tDCS application offers a potential adjuvant therapy for psychiatric disorders such as ADHD and may improve recovery of cognitive dysfunctions.

# Materials and methods

## Key resources table

| Reagent type (species) or resource | Designation | Source or reference | Identifiers | Additional information |
|---|---|---|---|---|
| Strain, strain background (Male. Rattus) | Wistar kyoto rats | Izm | | |
| Strain, strain background (Male. Rattus) | Spontaneously hypertensive rats | Izm | | |
| Chemical compound, drug | MPH | The United States Pharmacopeial Convention | 1433008 | 2 mg/kg |
| Chemical compound, drug | BrdU | Merck | B5002-5G | 50 mg/kg |
| Commercial assay or kit | TOPscriptTM cDNA Synthesis Kit | Enzynomics | EZ005S | Used following manufacturer's recommendations |
| Commercial assay or kit | Quantikine free BDNF ELISA kit | R and D Systems | DBD00 | Used following manufacturer's recommendations |
| Commercial assay or kit | TH ELISA kit | Cusabio | CSB-E13102r | Used following manufacturer's recommendations |
| Commercial assay or kit | Corticosterone ELISA kit | Enzo Life Sciences Inc | ADI-900–097 | Used following manufacturer's recommendations |
| Sequence-based reagent | *TH* forward | This paper | PCR primers | TTATGGTGCAGG GCTGCTGTCTT |
| Sequence-based reagent | *TH* reverse | This paper | PCR primers | ACAGGCTGGTAG GTTTGATCTTGG |
| Sequence-based reagent | *DAT* forward | This paper | PCR primers | CAGCCTATGGAA GGGAGTAAAG |
| Sequence-based reagent | *DAT* reverse | This paper | PCR primers | CACACTGAGGTA TGCTCTGATG |
| Sequence-based reagent | *VMAT2* forward | This paper | PCR primers | GCTCCTCACTA ACCCATTCATA |
| Sequence-based reagent | *VMAT2* reverse | This paper | PCR primers | GCTGGAGAAG GCAAACATAAC |
| Sequence-based reagent | *BDNF* forward | This paper | PCR primers | CCTGTGGAGG CTAAGTGGAG |
| Sequence-based reagent | *BDNF* reverse | This paper | PCR primers | CCTGCTCTGA AGGGTGCTT |
| Sequence-based reagent | *TGF-β1* forward | This paper | PCR primers | CTGTGGAGCA ACACGTAGAA |
| Sequence-based reagent | *TGF-β1* reverse | This paper | PCR primers | GTATTCCGTCT CCTTGGTTCAG |
| Sequence-based reagent | *GDNF* forward | This paper | PCR primers | CCAGAGAATTC CAGAGGGAAAG |
| Sequence-based reagent | *GDNF* reverse | This paper | PCR primers | CTTCACAGGAAC CGCTACAA |
| Sequence-based reagent | *NGF* forward | This paper | PCR primers | CCTCTTCGGACAATATGG |

*Continued on next page*

*Continued*

| Reagent type (species) or resource | Designation | Source or reference | Identifiers | Additional information |
|---|---|---|---|---|
| Sequence-based reagent | *NGF* reverse | This paper | PCR primers | CGTGGCTGTGGTCTTATC |
| Sequence-based reagent | *NT3* forward | This paper | PCR primers | AGGTCACAATTCCAGCCGAT |
| Sequence-based reagent | *NT3* reverse | This paper | PCR primers | GTTTCCTCCGTGGTGATGTT |
| Sequence-based reagent | *GAPDH* forward | This paper | PCR primers | TGAACGGGAAGCTCACTG |
| Sequence-based reagent | *GAPDH* reverse | This paper | PCR primers | GCTTCACCACCTTCTTGATG |
| Antibody | Anti- mBDNF (Rabbit polyclonal) | Novus | Cat#: NB100-98682 RRID:AB_1290643 | WB (1:1000) |
| Antibody | Anti-TGF-ß1 (Rabbit polyclonal) | Abcam | Cat#: ab92486 RRID:AB_10562492 | WB (1:500) |
| Antibody | Anti-GDNF (Mouse monoclonal) | Santa Cruz | Cat#: sc13147 RRID:AB_627672 | WB (1:500) |
| Antibody | Anti-NGF (Rabbit polyclonal) | Abcam | Cat#:ab6199 RRID:AB_2152414 | WB (1:500) |
| Antibody | Anti-NT3 (Rabbit polyclonal) | Alomne Labs | Cat#:ANT003 RRID:AB_2040013 | WB (1:200) |
| Antibody | Anti-TH (Rabbit polyclonal) | Abcam | Cat#:ab112 RRID:AB_297840 | WB (1:200) |
| Antibody | Anti-DAT (Rabbit polyclonal) | Millipore | Cat#:AB1591P RRID:AB_90808 | WB (1:100) |
| Antibody | Anti-VMAT2 (Rabbit polyclonal) | Novus | Cat#:NB110-68123 RRID:AB_1111327 | WB (1:500) |
| Antibody | Anti-mBDNF (Sheep polyclonal) | Abcam | Cat#:ab75040 RRID:AB_1280756 | IF (1:500) |
| Antibody | Anti-pTrkB (Rabbit polyclonal) | Abcam | Cat#:ab131483 RRID:AB_11156897 | IF (1:500) |
| Antibody | Anti-TH (Mouse monoclonal) | Santa Cruz | Cat#:sc25269 RRID:AB_628422 | IF (1:1000) |
| Antibody | Anti-DAT (Rat monoclonal) | Santa Cruz | Cat#:sc32259 RRID:AB_627402 | IF (1:50) |
| Antibody | Anti-BrdU (Rabbit monoclonal) | Bio-Rad | Cat#:MCA2483 RRID:AB_808349 | IF (1:500) |
| Antibody | Anti-NeuN (Rabbit polyclonal) | Millipore | Cat#:ABN78 RRID:AB_10807945 | IF (1:1000) |
| Antibody | Anti-Iba1 (Rabbit polyclonal) | Wako | Cat#:019–19741 RRID:AB_839504 | IF (1:500) |
| Software, algorithm | G*Power | G*Power 3.1 software | RRID:SCR_013726 | Germany; http://www.gpower.hhu.de/ |
| Software, algorithm | SigmaPlot | SigmaPlot 11.2 | RRID:SCR_010285 | |

*Continued on next page*

*Continued*

| Reagent type (species) or resource | Designation | Source or reference | Identifiers | Additional information |
|---|---|---|---|---|
| Software, algorithm | ITK-SNAP | ITK-SNAP v3.8.0 | RRID:SCR_002010 | www.itksnap.org |
| Software, algorithm | MVGC Multivariate Granger Causality Matlab Toolbox | Matlab 9.8 | RRID:SCR_015755 | |
| Software, algorithm | BioMesh3D | version 1.5 | RRID:SCR_009534 | www.wias-berlin.de |
| Software, algorithm | CoMet | CoMet | RRID:SCR_011925 | |
| Software, algorithm | SMART Video-tracking | SMART v3.0 video tracking system | RRID:SCR_002852 | |
| Software, algorithm | IMT i-Solution | IMT i-Solution Inc 10.1 | | |

## Experimental procedures

Male Wistar Kyoto rats (WKY/Izm; n = 25) and spontaneously hypertensive rats (SHR/Izm; n = 65) were obtained from the Inasa and Nakaizu Production Facility (Japan SLC, Inc, Kotoh-cho, Japan). Rats were randomized into eight experimental groups (n = 10/group); WKY (control strain), SHR (model of ADHD), sham (SHR or WKY rats receiving sham stimulation), tDCS-PFC (SHR or WKY rats stimulated with anodal HD-tDCS over the prefrontal cortex), tDCS-M1 (SHR rats stimulated with anodal HD-tDCS over the primary motor cortex), and MPH (SHR rats treated with methylphenidate; positive control). Three-dimensional (3D) numerical simulations were performed to identify the stimulated patterns of brain regions according to the electrode position and current (*Figure 6A*). Then, HD-tDCS treatment was performed once a day for 5 days, beginning 7 days after electrode implantation, followed by a rest interval of 2 days, and then another 5 days of treatment. Behavioral tests (n = 5–8/group) of cognitive function were performed according to the experimental schedule presented in *Figure 6C*. All behavioral experiments were performed by independent, blinded observers. Expression of dopaminergic neurotransmission and NTF genes was evaluated using real-time polymerase chain reaction (PCR) of tissues (n = 4–5/group) obtained from the prefrontal cortex, striatum, hippocampus, and substantia nigra/ventral tegmental area. The expression and secretion of dopaminergic neurotransmission and NTFs were also evaluated using a western blot (n = 5/group) and enzyme-linked immunosorbent assay (ELISA, n = 4–5/group). Lastly, immunohistochemical analyses were performed to detect the distribution of specific cells of dopaminergic neurotransmission factors and NTFs. All data analyses used the mean of three samples per animal and were conducted by investigators who were blinded to genotypes and treatments. All experiments were approved by the Pusan National University Animal Care and Use Committee and were performed in accordance with the National Institutes of Health Guidelines (PNU-2018–1932).

## Animals

A total number of 90 (25 WKY/Izm and 65 SHR/Izm) male rats (PND 25 ± 1, 100–140 g) were obtained from the Inasa and Nakaizu Production Facility (Japan SLC, Inc). The animal room was temperature-controlled (22 ± 1°C) and ventilated, and maintained on a 12 hr light-dark cycle (dark from 06:00 to 18:00 hr). The animals had free access to food and water. We minimized animal suffering and used the minimum number of animals necessary to assess significance. The sample size was determined based on the speed (cm/s) of the open field test after tDCS treatments in our previous results using G*Power 3.1 software (Heinrich-Heine-Universität, Düsseldorf, Germany; http://www.gpower.hhu.de/). Our previous results suggested an expected sample size for this research study of n = 5 for each six groups, a fixed effect, omnibus, one-way ANOVA with six groups, and an effect size $f = 0.8$, at $\alpha = 0.05$ and $\beta = 0.2$. Consequently, group sizes of $n > 5$ were used for the behavioral tests. Computer-generated randomization was conducted using SigmaPlot 11.2 (Systat Software Inc,

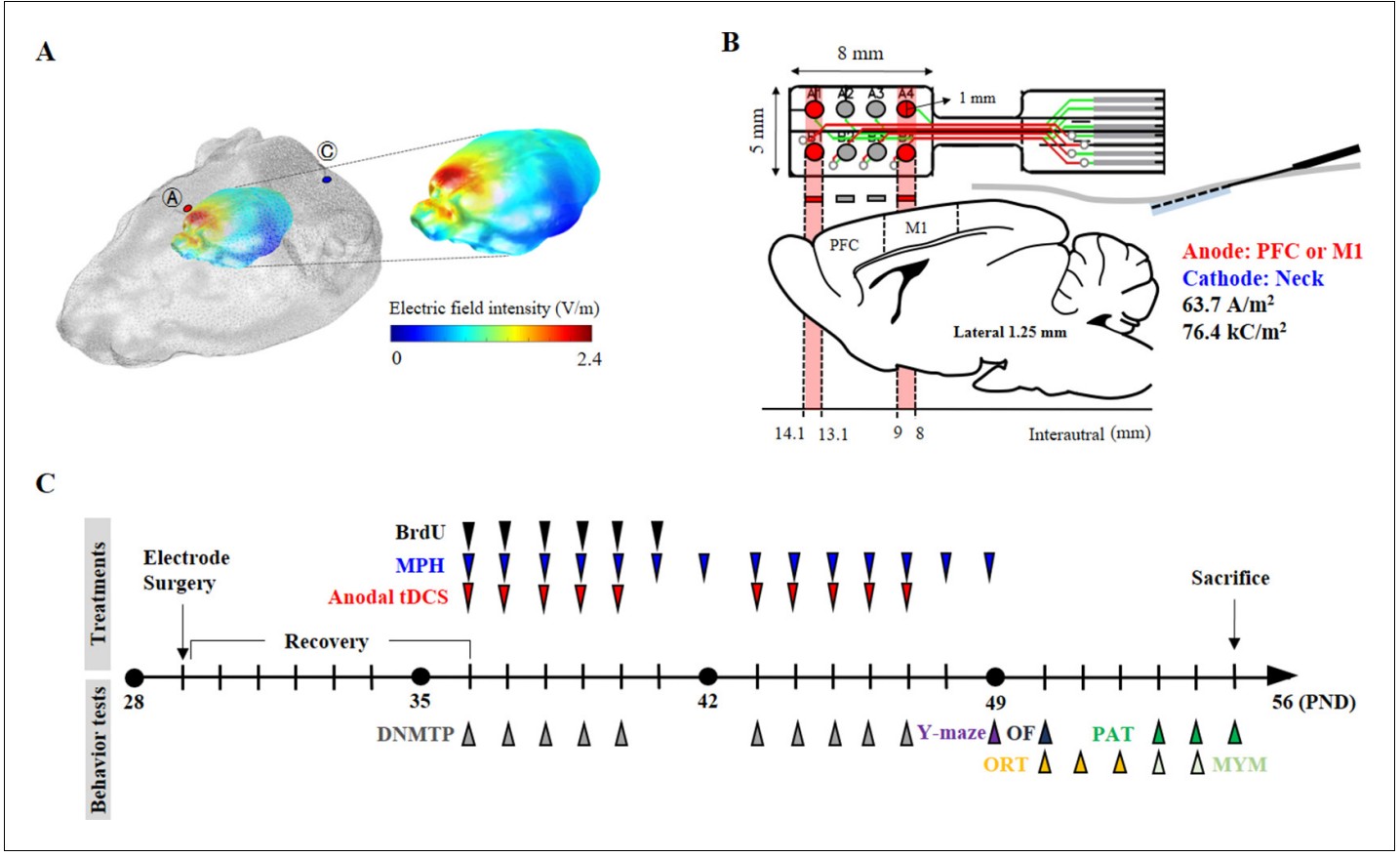

**Figure 6.** Experimental schematic diagram. (**A**) Three-dimensional tDCS simulation over the prefrontal cortex. The highest electric field intensity values are represented in red, and the lowest electric field intensity values are represented in blue. False color: electric field intensity (V/m). Ⓐ, anode electrode; ©, reference electrode. Anodal stimulation was delivered at an intensity of 63.7 A/m² over the prefrontal cortex. The predicted peak electric field intensity was 2.4 V/m in the frontal cortex. (**B**) Schematic diagram showing each electrode position and corresponding cerebral cortical region. The anodal stimulation was pointed over the frontal cortex and the primary motor cortex via the electrode, and the cathode was positioned on the skin of the neck. (**C**) The time schedule of HD-tDCS or MPH treatment and behavior tests. OF, open-field test; PAT, passive avoidance test; MYM, modified version of the Y-maze test.

The online version of this article includes the following figure supplement(s) for figure 6:

**Figure supplement 1.** Three-dimensional tDCS simulation over the prefrontal and primary motor cortex.

San Jose, CA, USA), to allocate 10 animals per groups to either sham or real treatment. After randomization, the SHR rats were allocated in a blinded fashion.

## Three-dimensional tDCS simulation

For the 3D tDCS simulation, magnetic resonance (MR) and microcomputed tomography (micro-CT) images from the head and neck of Long-Evans rats (NeuroImaging Tools and Resources Collaborator, https://www.nitrc.org) were employed. From the micro-CT image of a rat, a 3D skin and skull surface mesh model was generated using itk-SNAP (itk-SNAP v3.8.0, www.itksnap.org). Subsequently, a 3D surface mesh model from an MR image was generated. The three-surface mesh mode was configured as one 3D surface model mesh using Matlab 2020 (Matlab 9.8, c, Natick, MA, USA). The 3D volume mesh was generated using TetGen (version 1.5., www.wias-berlin.de) for finite element model (FEM) analysis. The 3D FEM model of the rat was comprised of four structures: scalp, skull, cerebrospinal fluid, and brain (conductivity properties: scalp, 0.465; skull, 0.015; cerebrospinal fluid, 1.65; and brain, 0.3). The generated volume mesh model was used for the 3D tDCS simulation using Comets Software (*Jung et al., 2013*; *Lee et al., 2017*). The 3D tDCS numerical simulation results predicted that a current at a density of 63.7 A/m² flowing from the anode electrode reaches the brain region close to the anode electrode. The placement of electrodes over the prefrontal or primary motor cortex predicted

that both peak electric field intensity and current density would be observed under the electrode and its periphery. The predicted peak electric field intensity was 2.4 V/m in the prefrontal cortex and 3.5 V/m in the primary motor cortex. The predicted peak current densities were 2.1 A/m$^2$ and 3.3 A/m$^2$ in the prefrontal cortex and primary motor cortex, respectively. In addition, it was also predicted that the areas where the electric field is thresholded above 2 V/m focalized to the prefrontal and primary motor cortex (*Figure 6A*, *Figure 6—figure supplement 1*).

## High-definition transcranial direct current stimulation

For the epicranial electrode implant, SHR and WKY rats were anesthetized using the VIP 3000 calibrated vaporizer (Midmark, Orchard Park, OH, USA) with 2% isoflurane (Choongwae, Seoul, Korea) on a heating pad (37°C). The scalp and underlying tissue were removed, and the electrode was implanted using dental cement (38216, Durelon carboxylate luting cement, 3M, Neuss, Germany). The anode at the center of the first two active electrodes (radius, 0.5 mm) was positioned over the prefrontal cortex (1.25 mm left and right, 4.35 mm anterior from bregma) or at the center of the last two electrodes were placed over the primary motor cortex (1.25 mm left and right, −0.5 mm from bregma) (*Figure 6B*). The non-epicranial electrode implant animals were also anesthetized at the same time for the surgical operation, and then all animals were transferred to care cases (under a heat lamp). After a check for generalized movements, they were moved to their home cage. An anodal HD-tDCS current was applied continuously for 20 min (intensity, 50 µA; current density, 63.7 A/m$^2$; charge density, 76.4 kC/m$^2$) using a constant current stimulator (8CH-tDCS; Neuro Rehap, Busan, Korea) under isoflurane anesthesia. The cathode was inserted below the skin of the back of the neck using a needle (0.30 mm in diameter and 40 mm in length) as an extracephalic reference electrode to avoid common side-effects including transient sensation at the skin (*Kronberg and Bikson, 2012*). The non-HD-tDCS groups were anesthetized at the same time for 20 min. HD-tDCS was repeated once a day beginning from 7 days after the electrode surgery, for a total of 5 consecutive days, followed by a rest period of 2 days before five additional daily treatments, to avoid possible stress induced by consecutive application. The current intensity was ramped for 30 s to prevent a stimulation break effect. The sham group was connected to the system for sham stimulation where no current was applied. On each day, the HD-tDCS intervention was performed at approximately the same time (at 10:00 a.m.). Pain or stress that may be caused by HD-tDCS application was estimated via measuring weight, resting times in the central zone in the open field test, and corticosterone levels in the plasma.

## Methylphenidate treatment

As a positive control, MPH (1433008, The United States Pharmacopeial Convention, North Bethesda, MD, USA) was injected once a day (i.p.; 2 mg/kg dissolved in saline) from 7 days after the electrode surgery, for 2 weeks. Other groups were injected with a 0.9% saline vehicle.

## BrdU labeling

To label proliferating cells in the brain, animals received BrdU (i.p.; 50 mg/kg; B5002-5G, Merck KGaA, Darmstadt, Germany) once a day for 6 consecutive days beginning 7 days after the electrode surgery. BrdU is a synthetic thymidine analog that is incorporated into the DNA of dividing cells during the S-phase and can subsequently be detected with an anti-BrdU-specific antibody.

## Open-field test

Rats were assessed in an open field apparatus to evaluate spontaneous locomotor behavior. The open field (60 × 60 × 30 cm$^3$) was a black square box divided into a center (30 × 30 cm$^2$) and a surrounding zone. Rats were placed in an individual box in the center, following 5 min of habituation. During the 15 min test, the total distance (cm), velocity (cm/s), and resting times in the center zone (s) were measured using a SMART v3.0 video tracking system (Panlab, S.L.U., Barcelona, Spain), under low-light intensity (<50 lx) in a quiet room.

## T-maze, delayed non-match to place version

The DNMTP version of the T-maze was used to assess the animals' spatial working memory using rewarded alternation. Rats underwent 2 days of habituation, during which they were allowed to

freely visit all arms of the T-maze for 10 min. During a sample phase, rats were placed in the starting arm and forced to randomly enter either the right or left arm, receiving a food reward (*Figure 1— figure supplement 1B*). For the test phase, both arms were open, and there was one food reward arm and one empty (error) arm. The intra-trial retention interval (delay) between the forced run and the choice run was set at 10 s, and the inter-trial interval between trial pairs was 30 min. All animals performed more than seven trials in each session; the index of reward performance was calculated as the average of the correct number of choices.

## Y-maze, modified and spontaneous alternation task version

A modified version of the Y-maze test was used to assess working memory and behavioral flexibility. The apparatus used consisted of black polypropylene walls with three arms, each 40 cm long, 10 cm wide, and 16 cm high. Rats were habituated for 2 days before testing. In the sample phase trial, each animal was individually placed in the maze with one of the three arms closed. The animals were allowed to explore the other two arms freely for 10 min. The test phase trial was conducted 24 hr after the sample phase trial. The previously closed arm was opened in the test phase trial and defined as the 'new arm'. Behavior was video-recorded for later analysis. Percent time spent in the new arm was measured using a SMART v3.0 video tracking system (Panlab, S.L.U., Barcelona, Spain) in a quiet room.

Y-maze spontaneous alternation is a test used to measure spatial working memory during exploratory activity. Rats freely explored the three arms of a Y-shaped maze for 30 min to habituate in the maze, and the task began at the center of the maze 10 min after habituation. Each rat was placed at the end of one arm (labeled 'A', 'B', or 'C') and allowed to move freely through the maze for 8 min. SAP was defined as visiting three different arms consecutively (i.e. ABC, ACB, BCA, BAC). Percent spontaneous alternation was calculated as [(number of alternations) / (total arm entries − 2)]×100. The AARs and SARs were also scored in order to assess aspects of attention within spontaneous exploratory behavior.

## Passive avoidance test

The passive avoidance test assesses learning and memory. The passive avoidance test used here (Med Associates Inc, St. Albans, VT, USA) consisted of three sessions. An illuminated compartment ($20 \times 20 \times 25$ cm$^3$) and a dark compartment ($20 \times 20 \times 25$ cm$^2$) were separated by a sliding door. In the first session, each rat underwent training trials moving from the light compartment to the dark compartment. After training, when the rats entered the dark compartment, the door closed automatically and a single inescapable scrambled foot shock (0.8 mA; 3 s) was delivered through the grid floor. During the final session, the procedure was repeated until the latency to enter the dark box was ≥300 s.

## Object-place recognition test

For the object-place recognition test to evaluate spatial memory with discrimination, rats were individually habituated to an open-field box ($60 \times 60 \times 30$ cm$^3$) for 10 min. After habituation, the rats were allowed to explore two identical objects (each of which was a $5 \times 5 \times 15$ cm$^3$ bottle; referred to as 'A1' and 'A2') for 10 min, positioned at the back corners of the arena, 10 cm from the wall. The total time that the rats spent exploring each of the two objects was measured, and then the rat was returned to the home cage. In a test trial performed 24 hr later, one of the objects was displaced to a new position (B1) while the other object (A2) remained at the same location. Novel place discrimination was assessed by comparing the time spent exploring the familiar object (A2) with that exploring the novel object (B1). Data were used to determine a discrimination score using the following equation: [(B1 − A2)/ (B1 + A2)]. The time spent exploring each object was analyzed using a SMART v3.0 video tracking system (Panlab, S.L.U., Barcelona, Spain).

## Quantitative real-time PCR

Total RNA was isolated from frozen brain tissue from the prefrontal cortex, striatum, hippocampus, and SN/VTA, using TRIzol reagent (Thermo Fisher Scientific, Waltham, MA, USA). Extracted RNA was dissolved in diethyl pyrocarbonate-treated water. The purity and integrity of extracted RNA were evaluated by optical density measurements (260/280 nm ratios) using a Nanodrop

Spectrophotometer (nd-1000, Thermo Fisher Scientific). Total RNA was reverse transcribed into cDNA using a TOPscriptTM cDNA Synthesis Kit (EZ005S, Enzynomics, Daejeon, Korea). Then, 2 mg of cDNA and gene-specific primers were added to SYBR green mastermix (RR420A, Takara Bio Inc, Kusatsu, Japan), and subjected to PCR amplification. Primer sets are listed below:

> *TH* forward, 5′- TTATGGTGCAGGGCTGCTGTCTT −3′
> *TH* reverse, 5′- ACAGGCTGGTAGGTTTGATCTTGG −3′
> *DAT* forward, 5′- CAGCCTATGGAAGGGAGTAAAG −3′
> *DAT* reverse, 5′- CACACTGAGGTATGCTCTGATG −3′
> *VMAT2* forward, 5′- GCTCCTCACTAACCCATTCATA −3′
> *VMAT2* reverse, 5′- GCTGGAGAAGGCAAACATAAC −3′
> *BDNF* forward, 5′- CCTGTGGAGGCTAAGTGGAG −3′
> *BDNF* reverse, 5′- CCTGCTCTGAAGGGTGCTT −3′
> *TGF- β1* forward, 5′- CTGTGGAGCAACACGTAGAA −3′
> *TGF- β1* reverse, 5′- GTATTCCGTCTCCTTGGTTCAG −3′
> *GDNF* forward, 5′- CCAGAGAATTCCAGAGGGAAAG −3′
> *GDNF* reverse, 5′- CTTCACAGGAACCGCTACAA −3′
> *NGF* forward, 5′- CCTCTTCGGACAATATGG −3′
> *NGF* reverse, 5′- CGTGGCTGTGGTCTTATC −3′
> *NT3* forward, 5′- AGGTCACAATTCCAGCCGAT −3′
> *NT3* reverse, 5′- GTTTCCTCCGTGGTGATGTT −3′
> *GAPDH* forward, 5′- TGAACGGGAAGCTCACTG −3′
> *GAPDH* reverse, 5′- GCTTCACCACCTTCTTGATG −3′

Denaturation was performed at 95℃ for 10 min, after which, segment 1 (95 ℃ for 10 s), segment 2 (60 ℃ for 15 s), and segment 3 (72 ℃ for 10 s) were repeated for 40 cycles. Relative expression data were quantified using $2^{-\Delta\Delta CT}$, where the threshold value (Ct) is the threshold value. Glyceraldehyde-3-phosphate dehydrogenase (GAPDH) was used as a housekeeping gene. Gene expressions were expressed as fold changes of WKY.

## Western blotting

Frozen tissue was homogenized in a lysis buffer containing 250 mM NaCl, 5 mM EDTA (pH 7.5), 25 mM Tris-HCl (pH 7.5), 1% NP40, 1 mM phenylmethylsulfonyl fluoride (PMSF), 5 mM dithiothreitol (DTT), 0.1 mM sodium orthovanadate ($Na_3VO_4$), 10 mM NaF, leupeptin, and protein inhibitor cocktail at 4℃ for 5 min. Proteins from the soluble fraction (20 μl) were resolved using 10% sodium dodecyl sulfate-polyacrylamide gel electrophoresis (SDS-PAGE) and then transferred to a nitrocellulose membrane (Whatman, Piscataway, NJ, USA) using a blotting system (Bio-Rad). The membrane was incubated in a 5% solution of non-fat milk for 1 hr at 20℃. The membrane was then washed and incubated overnight at 4 ℃ with primary antibodies against mBDNF (NB100-98682, Novus Biologicals, LLC., Centennial, CO, USA), TGF-ß1 (ab92486, Abcam, Cambridge, UK), GDNF (sc13147, Santa Cruz Biotechnology, Inc, Dallas, TX, USA), NGF (ab6199, Abcam), NT3 (ANT003, Alomne Labs, Jerusalem, Israel), TH (ab112, Abcam), DAT (AB1591P, Millipore Corporation, Billerica, MA, USA), and VMAT2 (NB110-68123, Novus). The blots were washed for 20 min and incubated at room temperature for 1 hr with secondary antibodies according to the appropriate species: anti-mouse or anti-rabbit immunoglobulin G (ADI-SAB-100 or 300; Enzo Life Sciences, Inc, Farmingdale, NY, USA). Quantification of band immunoreactivity was visualized using an enhanced chemiluminescence system (Pierce, Rockford, IL, USA) and imaged with an Image Quant LAS-4000 imaging system (Fujifilm, Tokyo, Japan). Western blotting was analyzed using the IMT i-Solution Inc 10.1 (17TH-5989 Walter Gage Rd., Vancouver, BC, CA) image analysis software. Protein expressions/ß-actin ratios were calculated, and data were expressed as percentages of the mean values of WKY protein levels.

## Enzyme-linked immunosorbent assay (ELISA)

Protein levels for mBDNF, TH, and corticosterone were measured using a Quantikine free BDNF ELISA kit (DBD00, R and D Systems Inc, Minneapolis, MN, USA), a TH ELISA kit (CSB-E13102r, Cusabio Biotech CO., LTD, Wuhan, China), and a corticosterone ELISA kit (ADI-900–097, Enzo Life Sciences Inc, Farmingdale, NY, USA). Tissue samples for free BDNF ELISA were homogenized with lysis buffer containing 200 mM Tris (pH 8.0), 150 mM NaCl, 2 mM EDTA, 1 mM NaF, 1% NP40, 1 mM PMSF, 1 mM Na3VO4, and a protease inhibitor cocktail, and then centrifuged at 13,000 rpm for 30

min at 4°C. TH ELISA samples were obtained from 30 mg of tissue homogenized with 150 µl of 1 X PBS and stored overnight at −20°C. After two freeze-thaw cycles, samples were centrifuged at 5000 rpm for 5 min at 4°C. For corticosterone ELISA, blood samples were collected from the heart within 3 min using a needle and transferred to a EDTA BD Vacutainer containing blood collections tubes (367835, Becton Dickinson, Franklin Lakes, NJ, USA). The plasma was separated from the whole blood by centrifuging at 5000 g for 5 min at 4°C. Samples and standards were applied to titer plates. Final reactions were measured using a SpectraMax 190 microplate reader (Molecular Devices, Sunnyvale, CA, USA). A standard curve of recombinant BDNF (0–4,000 pg/well), TH (0–5,000 pg/well), and corticosterone (0–20,000 pg/well) was plotted for each plate. The average value of the sample was normalized against the total protein concentration.

## Immunofluorescence staining

For immunofluorescence analysis, 25 µm serial cryosections were obtained using a CM3050 cryostat (Leica Microsystems, Wetzlar, Germany). After incubation in a blocking buffer (1 × PBS/1% bovine serum albumin/0.3% Triton X-100) for 1 hr at room temperature, sections were incubated with the following primary antibodies overnight in PBS at 4°C: mBDNF (ab75040, Abcam), pTrkB (ab131483, Abcam), TH (sc25269, Santa Cruz Biotechnology, Inc), DAT (sc32259, Santa Cruz Biotechnology, Inc), BrdU (MCA2483, Bio-Rad, Hercules, CA, USA), NeuN (ABN78, Millipore Corporation), and Iba1 (019–19741, Wako Chemicals, Richmond, VA, USA). Sections were washed with PBST and incubated with fluorescent secondary antibodies (A11001, A11007, or A11037, Invitrogen, Carlsbad, CA, USA; CA94010, Vector Laboratories, Inc, Burlingame, CA, USA) for 2 hr at room temperature. Sections were mounted onto slides using a mounting medium (H-1200, Vector Laboratories, Inc) and imaged using a fluorescence microscope (Carl Zeiss Imager M1, Carl Zeiss AG, Oberkochen, Germany). Immunofluorescence was analyzed using the IMT i-Solution Inc 10.1 (17TH-5989 Walter Gage Rd., Vancouver, BC, CA) image analysis software.

## Statistical analyses

All data are expressed as the mean ± standard error of the mean (SEM). Data were analyzed using SigmaPlot Version 11.0 (Systat Software, San Jose, CA, USA). Statistical analyses were performed using one-way analyses of variance (ANOVAs) for repeated measures, followed by Tukey's tests for multiple comparisons. $p < 0.05$ was considered a significant difference. All statistical analyses were performed on raw data. No samples were excluded from analysis.

## Additional information

### Funding

| Funder | Grant reference number | Author |
|---|---|---|
| National Research Foundation of Korea | 2018R1A2A2A05018926 | Byung Tae Choi |
| National Research Foundation of Korea | 2014R1A5A2009936 | Byung Tae Choi |

The funders had no role in study design, data collection and interpretation, or the decision to submit the work for publication.

### Author contributions

Da Hee Jung, Conceptualization, Data curation, Formal analysis, Methodology, Writing - original draft; Sung Min Ahn, Malk Eun Pak, Hong Ju Lee, Methodology; Young Jin Jung, Investigation; Ki Bong Kim, Yong-Il Shin, Data curation; Hwa Kyoung Shin, Data curation, Methodology; Byung Tae Choi, Conceptualization, Data curation, Formal analysis, Investigation, Writing - original draft

### Author ORCIDs

Ki Bong Kim https://orcid.org/0000-0001-5724-4653
Byung Tae Choi https://orcid.org/0000-0002-5965-4346

### Ethics

Animal experimentation: All experiments were approved by the Pusan National University Animal Care and Use Committee and were performed in accordance with the National Institutes of Health Guidelines (PNU-2018-1932).

### Decision letter and Author response

Decision letter https://doi.org/10.7554/eLife.56359.sa1
Author response https://doi.org/10.7554/eLife.56359.sa2

## Additional files

### Supplementary files

• Transparent reporting form

### Data availability

All data generated or analysed during this study are included in the manuscript and supporting files. Source data files have been provided for Figures and Tables.

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
