## [Decision Letter]

**Acceptance summary:**

Transcranial direct current stimulation (tDCS) is among the most investigated interventional neurotechnolgies in cognitive neuroscience and for neuropsychiatric indictions. This work adds rigor through using high-definition tDCS in a rat model of ADHD, showing alleviation cognitive deficits linked to dopaminergic neurotransmission factors, increased expression of several neurotrophic factors involving BDNF, and activated hippocampal neurogenesis. This work supports the notion that tDCS can activated a consultation of linked neuro-restorative processes, with broad applications.

**Decision letter after peer review:**

Thank you for submitting your article "Therapeutic effects of anodal transcranial direct current stimulation in a rat model of ADHD" for consideration by *eLife*. Your article has been reviewed by three peer reviewers, one of whom is a member of our Board of Reviewing Editors, and the evaluation has been overseen by Richard Ivry as the Senior Editor. The reviewers have opted to remain anonymous.

The reviewers have discussed the reviews with one another and the Reviewing Editor has drafted this decision to help you prepare a revised submission.

As the editors have judged that your manuscript is of interest, but as described below substantial changes are required to determine its suitability for publication.

We would like to draw your attention to changes in our revision policy that we have made in response to COVID-19 (https://elifesciences.org/articles/57162). Because many researchers have temporarily lost access to the labs, we will give authors as much time as they need to submit revised manuscripts. We are also offering, if you choose, to post the manuscript to bioRxiv (if it is not already there) along with this decision letter and a formal designation that the manuscript is 'in revision at *eLife*'. Please let us know if you would like to pursue this option. (If your work is more suitable for medRxiv, you will need to post the preprint yourself, as the mechanisms for us to do so are still in development.)

Summary:

This is an intriguing study that bridges molecules to behavior in establishing a mechanism and rational for the use of tDCS as a potential intervention in ADHD. These results indicate that tDCS can alleviate behavioral deficits in a rat model of ADHD. The study appears to be very thorough, both in the battery of behavioral tests used and assays used to measure changes in dopaminergic pathways, neurotrophic pathways and hippocampal neurogenesis.

Major issues for revision:

1) Important methodological details are missing or unclear. This concern must be fully addressed to warrant review of a revision. To provide a few examples:

a) Lack of clarity on dosing (unacceptable to refer reader to prior publications on critical methodological details). Indeed, if the dosing is ultimately not so focal then the current is passing through most of the brain, brain stem, spinal cord, periphery etc. This would change the potential impact of the paper.

b) Concerns about control/sham are equally essential. If the stimulation provided (electric field in brain) is much higher than tDCS, the paper may not be suitable for publication without additional experiments.

c) What were the animals doing during the tDCS session?

d) What time of day was it applied?

e) The anodes are not explained well and the cathodes completely opaque. The current density is presumably quite high for the anodes, but might be even much higher for the cathodes, which are presented as needles.

f) There is no data that present whether the rats are comfortable with the stimulation or not. Pain and stress might lower movement or enhance agitation in rats, depending on the strain and individual biological variability.

g) The lack of information on a reasonable control groups is also the reason that there is no data about the effects of the surgical procedure and the post-operative care.

2) Issues with sham. Procedure for sham is not clear. Is it no stimulation or a ramp up/down? What sensation does it produce? An "active control" would have some advantages over sham and may be essential if the sham is problematic.

3) As understood from the text, the Wistar-Kyoto rats (WKY), which are indeed the best control for SHR rats, were not operated at all, and they were completely naïve. There is no WKY-sham group, and no WKY-tDCS group. The lack of proper control groups makes the drawing of conclusions difficult.

4) Stimulation intensity. There is no discussion of the choice of current intensity or how this intensity is related to human tDCS techniques. The authors report current/charge density at the skull which is close to values that created lesions in other studies (Jackson et al., 2017). Of course, it is unlikely that lesions were created here, but this suggests that the electric field/current density generated in the rat brain may be very high. In the absence of evidence that similar behavioral effects are observed with weaker stimulation, or evidence that the stimulation intensity used here is comparable to human tDCS, the relevance of this study for treating ADHD in humans is limited. A good way to address this issue is a computational model as in Jackson et al., 2017. Otherwise, the authors may be able to extrapolate from other models or intracranial measurements to approximate the electric field magnitude in their study. tDCS is low intensity. If this paper is high intensity, maybe even supra-threshold intensity, it highly limited (as much proper animal tDCS work) and the authors should not resubmit without more experiments.

5) The effects of such chronic stimulation on the tissue below the anodes was not investigated with histopathological studies. The tissue below the electrodes might suffer some damage, though previous studies showed that such current density does not inflict tissue damage, it is worthwhile to examine that with these unique electrodes.

6) A potential confound in animal studies of electrical stimulation is the role of somatosensation or peripheral nerve stimulation due to electrodes placed on the body. A control condition with active stimulation that does not target relevant brain areas (or the brain at all) can help alleviate this issue. To this end, comparisons between tDCS-PFC and tDCS-M1 are useful. It appears that in many of the behavioral assays used here, both stimulation types have beneficial effects. To what degree could the common effects of these protocols be explained by common peripheral effects? The authors should discuss this potential confound.

7) The authors claim that they perform an HD-tDCS. The montage shown is not HD-tDCS per se but a version modified for animals. This could be innovative but need to know more about dosing and brain current flow. This montage probably makes most of the electricity to run via the skin on the back of the neck.

8) This work has the potential to be of high translational impact. However, it is not clear how the stimulation intensity used here relates to studies in humans, which is critical for its translational impact, and therefore its suitability for publication in *eLife*. Can this dosing be scaled to human dosing?

9) While some results separate by dose, others do not, and does this question focality? tDCS of M1 showed similar effects to those of the PFC group in both reduction in hyperactivity component (Results paragraph one and Figure 1A) and in passive avoidance test (Figure 1E). This might be due to spillage of current.

10) The effects of the surgical procedure and the post-operative measures were not investigated as a factor. The stress of the surgical procedure and the restraint required for proper sham or tDCS application and the inconvenience/pain that might be due to the electricity in the real tDCS might be a factor that was not investigated. For example, the timing of the biochemical sampling of TH and BDNF levels is not necessarily related to tDCS beneficial effects. The sampling was 8 days after the last stimulation. Whether this is related to behavior or stress is really not answered. Reviewers had mixed views on if it would be better/needed if the WKY would also contain sham and tDCS, on the expense of the M1 group for example. The positive control group of the MPH is useful.

[Editors' note: further revisions were suggested prior to acceptance, as described below.]

Thank you for resubmitting your work entitled "Therapeutic Effects of Anodal Transcranial Direct Current Stimulation in a Rat Model of ADHD" for further consideration by *eLife*. Your revised article has been evaluated by Richard Ivry (Senior Editor) and Marom Bikson (Reviewing Editor).

The manuscript has been improved but there are some remaining issues that need to be addressed.

1) The original concern regarding stimulation intensity and focality remains. An essential question that remains unanswered is how the brain electric field distribution in this study compares to typical human tDCS (or what is achievable with human tDCS). While the authors have created a model as recommended, it is not clear how their use of the model addresses this fundamental issue. The model should address 1) whether the stimulation produces too strong of an electric field in the brain to be comparable to human tDCS (intensity) and 2) the degree to which this stimulation is focal (e.g. restricted to PFC or M1). Instead the authors seem to use the modeling results to show that brain current density peaks under the anode and moves when the anode is moved. These two points are mostly trivial (they have been noted by many other studies). Moreover, the way that the model is presented (Figure 6) does not facilitate any quantitative comparison with human tDCS or other animal mechanistic studies. Some suggestions for how to improve the presentation and discussion of the model results.

a) Report electric field: the majority of mechanistic or neurophysiological studies of electrical stimulation have controlled for electric field magnitude. The modeling results should be reported in these units to facilitate comparison across systems.

b) Peak electric field: We know what peak electric fields in the human cortex during tDCS are (~1 V/m). How does the peak electric field compare in this study? The peak electric field here may be an order of magnitude higher.

c) Focality: several studies have suggested that measurable effects on neuronal firing require ~1V/m electric field. The model can help predict how much of the brain crosses this pseudo-threshold in your study. Is it restricted to PFC/M1? Does it reach subcortical structures? It may be that most of the brain in this study is above or near this pseudo-threshold.

d) Colormap: Consider using a colormap where electric field magnitude corresponds to color intensity and perhaps an additional figure where the electric field is thresholded above 1 V/m, making the degree of focality obvious.

These changes would help facilitate comparison with human tDCS, and help other researchers determine whether these results are likely to apply to their protocol. They may also help interpret your own results (e.g. perhaps PFC and M1 stimulation exhibit similar effects due to direct electrical stimulation of subcortical structures with both montages).

2) With regard to modeling technique, you should explicitly model CSF with a different conductivity from the brain itself. CSF strongly affects current flow patterns due to its high conductivity.

3) The western blotting has been removed altogether from the manuscript in response to the comment that BDNF was not in the size that fits the mature form. The authors' claim that they cannot display the western blots according to the journal requirements-the argument for this is not clear. Please explain. And if removed from the results, then reference to them should be removed from the Materials and methods, too.

4) The authors depend on ELISA for measuring BDNF. The assay used is actually for a human BDNF. It is true that there is ~50% cross-reactivity, but is there good justification that this is a reliable BDNF assay for the rat?

---

## [Author Response]

Major issues for revision:1) Important methodological details are missing or unclear. This concern must be fully addressed to warrant review of a revision. To provide a few examples:a) Lack of clarity on dosing (unacceptable to refer reader to prior publications on critical methodological details). Indeed, if the dosing is ultimately not so focal then the current is passing through most of the brain, brain stem, spinal cord, periphery etc. This would change the potential impact of the paper.

We appreciate your helpful comment. We additionally performed three-dimensional (3D) numerical tDCS simulations to identify the stimulating patterns of brain regions according to the electrode position and proper dosing of tDCS treatment. As for your comment, the major concerns associated with tDCS treatment are related to its safety and cortical excitability. Based on the 3D tDCS simulation results, we identified that a current of 63.7 A/m^2^ flowing from the anode electrode penetrates the skull and reaches the brain region close to the anode electrode. In addition, our results predicted that as the position of the anode changes, the position of the brain region that is intensively stimulated also changes. They also predicted no brain lesions (predicted brain injury range 6.3–13 A/m^2^) (Bikson et al., 2016; Antal et al., 2017).

b) Concerns about control/sham are equally essential. If the stimulation provided (electric field in brain) is much higher than tDCS, the paper may not be suitable for publication without additional experiments.

In the previously submitted manuscript, only SHR sham group was employed (model of ADHD). We performed additional experiments and compared parameters between sham and tDCS treatment groups among WKY rats (genetic control strain) (Figure 1-figure supplement 2). We also confirmed the safety of tDCS through corticosterone levels (Figure 2—figure supplement 2) and microglial activation (Figure 3—figure supplement 1). Additionally, we confirmed the animals’ weight during the HD-tDCS sessions, which showed no difference between the control, sham, and tDCS groups. Moreover, we added a clearer description of the sham groups to the Materials and methods section.

c) What were the animals doing during the tDCS session?

Anodal tDCS current was applied continuously for 20 min under isoflurane anesthesia. Animals in the non-tDCS group underwent the same procedure as the stimulated groups for 20 min, without any current being applied.

d) What time of day was it applied?

We performed the tDCS application at the same time daily (at 10:00 a.m.).

e) The anodes are not explained well and the cathodes completely opaque. The current density is presumably quite high for the anodes, but might be even much higher for the cathodes, which are presented as needles.

We added a clearer and more detailed description of the anode and cathode to the Materials and methods section. Cathode stimulation was applied to the back of the necks, via extracephalic reference electrodes, to exclude an effect of the reference electrode. Not only the surface electrode but also the needle electrode can be used as a tDCS reference electrode (Morya et al., 2019, PMID 31730494). Moreover, the standard pad electrode led to a higher concentration of current density at the pad electrode edges, and we employed needle cathode inserting below the skin of the back of the neck to avoid common side-effects including transient sensation at the skin (Kamida et al., 2011, PMID 20826186; Morya et al., 2019, PMID 31730494; Kronberg and Bikso, 2012).

f) There is no data that present whether the rats are comfortable with the stimulation or not. Pain and stress might lower movement or enhance agitation in rats, depending on the strain and individual biological variability.

We agree with your comment and performed additional experiments on the resting times in the open field test, weight during tDCS treatment, and corticosterone levels (Figure 2-figure supplement 2) in the plasma to confirm pain and stress levels caused by the tDCS stimulation. All these additional results show that tDCS did not induce pain and stress in the animals.

g) The lack of information on a reasonable control groups is also the reason that there is no data about the effects of the surgical procedure and the post-operative care.

We added a sentence about the control groups for the surgical procedure and the post-operative care. All animals including the control groups were anesthetized at the same time (for 20 min) and then transferred to care cases (under a heat lamp). After checking for generalized movements, they were moved to their home cage.

2) Issues with sham. Procedure for sham is not clear. Is it no stimulation or a ramp up/down? What sensation does it produce? An "active control" would have some advantages over sham and may be essential if the sham is problematic.

Following your suggestion, we included an additional sentence on the sham treatments. The current intensity was ramped for 30 s to prevent a stimulation break effect. The sham group was connected to the system for sham stimulation, while no current was applied. Moreover, we performed additional experiments and also compared between the sham and tDCS treatment groups among the genetic control strain WKY.

3) As understood from the text, the Wistar-Kyoto rats (WKY), which are indeed the best control for SHR rats, were not operated at all, and they were completely naïve. There is no WKY-sham group, and no WKY-tDCS group. The lack of proper control groups makes the drawing of conclusions difficult.

Following the reviewer’s comments, we performed additional experiments on behaviors and ELISA analysis in WKY, WKY-sham, and WKY-tDCS-PFC groups to confirm that WKY is the ideal control for SHR rats. We measured the concentration of TH and BDNF in the plasma, along with cognitive behaviors and hyperactivity for these groups. There were no significant differences among the WKY, WKY-sham, and WKY-tDCS-PFC groups. Interestingly, no behavioral changes and no changes in levels of TH and BDNF were observed in the WKY groups, indicating that WKY rats are a good control for our SHR ADHD animal model.

4) Stimulation intensity. There is no discussion of the choice of current intensity or how this intensity is related to human tDCS techniques. The authors report current/charge density at the skull which is close to values that created lesions in other studies (Jackson et al., 2017). Of course it is unlikely that lesions were created here, but this suggests that the electric field/current density generated in the rat brain may be very high. In the absence of evidence that similar behavioral effects are observed with weaker stimulation, or evidence that the stimulation intensity used here is comparable to human tDCS, the relevance of this study for treating ADHD in humans is limited. A good way to address this issue is a computational model as in Jackson et al., 2017. Otherwise, the authors may be able to extrapolate from other models or intracranial measurements to approximate the electric field magnitude in their study. tDCS is low intensity. If this paper is high intensity, maybe even supra-threshold intensity, it highly limited (as much proper animal tDCS work) and the authors should not resubmit without more experiments.

We agree that our tDCS charge density was higher than that used in clinical studies with human patients. For our study design, we needed to assess the safety of tDCS and cortical excitability in rodents. We referred to the previous literature, such as to studies by Liebetanz et al., 2009, and Jackson et al., 2017, to assess possible brain damage according to anode intensity. We also confirmed that previous studies used similar current and charge densities (63.7 A/m^2^, 76.4 kC/m^2^). Our employed densities are very close to values that create lesions, but the smaller the electrode the lower the lesion area in the same current density; the lesion area is thus dependent on the size of the electrode (Jackson et al., 2017). We thus used a smaller electrode than previous studies (0.785 mm^2^, radius 0.5 mm) (Jackson et al., 2017).

Additionally, we confirmed the lesion area by H&E staining. The PFC of the tDCS-PFC group and the M1 of the tDCS-M1 group did not show lesions, indicating that this current density does not induce tissue damage).

However, this density is two orders of magnitude higher than the charge density currently applied in clinical studies with human patients (ranging from 170 to 480 C/m^2^) (Liebetanz et al., 2009). Thus, effects cannot simply be translated into the clinics, which is a problematic issue for most experimental tDCS-studies. This is now mentioned in the Discussion section, with three additional references.

Moreover, following your suggestions, we additionally performed three-dimensional (3D) numerical simulations to predict the safety of our tDCS approach and the induced cortical excitability. Based on our 3D tDCS simulation results, a 50-uA current (63.7 A/m^2^) flowing from the anode electrode penetrates the skull and reaches the brain regions; especially those close to the anode electrode are intensively stimulated. If the current is increased on the anode electrode at the same anatomical position and shape, the stimulated pattern on the rat cortex proportionally increases. In addition, our results predicted that as the position of the anode changes, the position of the brain region that is intensively stimulated also changes. When the anode located in the PFC or M1 and the cathode located at the neck of the animal were stimulated with 63.7 A/m^2^, the peak of the electrode field of the value of the HD-tDCS was 2.0 and 3.2 A/m^2^ in the PFC and M1, respectively. No brain lesions were predicted, because animal models indicate that brain injury occurs at 6.3–13 A/m^2^ (Bikson et al., 2016; Antal et al., 2017).

5) The effects of such chronic stimulation on the tissue below the anodes was not investigated with histopathological studies. The tissue below the electrodes might suffer some damage, though previous studies showed that such current density does not inflict tissue damage, it is worthwhile to examine that with these unique electrodes.

Following your suggestion, we performed additional investigations for histopathological analysis through HE and Iba-1 staining. The results did not show brain damage due to histological and microbial changes. Moreover, microglial activation by Iba-1 staining was significantly reduced in the SHR-tDCS group compared to that in the SHR group in the prefrontal cortex, in line with previous studies showing that tDCS reduces inflammation.

6) A potential confound in animal studies of electrical stimulation is the role of somatosensation or peripheral nerve stimulation due to electrodes placed on the body. A control condition with active stimulation that does not target relevant brain areas (or the brain at all) can help alleviate this issue. To this end, comparisons between tDCS-PFC and tDCS-M1 are useful. It appears that in many of the behavioral assays used here, both stimulation types have beneficial effects. To what degree could the common effects of these protocols be explained by common peripheral effects? The authors should discuss this potential confound.

We fully agree with your comment. This may be an important limitation of this study, in that the efficacy of and the mechanism underlying tDCS have not been fully elucidated. As seen in our 3D tDCS analysis, tDCS application is predicted to stimulate certain cerebral regions. However, the mechanisms underpinning tDCS findings are often controversial, as studies show that the current reaching the cortex may not be strong enough to entrain neural activity (much of the current applied through a scalp electrode never reaches the target cortex). Some previous studies report that the efficacy of tDCS may be affected by transcutaneous stimulation of peripheral nerves or somatosensation in the skin (Tsuiki et al., 2019; Asamoah et al., 2019). However, we suggest that the main cause of these effects may be a connectivity-based spread from the prefrontal cortex to subcortical sites via mediating NMDA receptors, as mentioned in the Discussion section of our manuscript. We now fully describe this issue in the Discussion section.

7) The authors claim that they perform an HD-tDCS. The montage shown is not HD-tDCS per se but a version modified for animals. This could be innovative but need to know more about dosing and brain current flow. This montage probably makes most of the electricity to run via the skin on the back of the neck.

We agree that our montage is not HD-tDCS per se but a modified version. However, high-definition (HD) electrodes, not pad electrodes, are circular, <1 cm in diameter, and employ a higher number and density of HD electrodes. When one or more HD electrodes are used, tDCS is called high-definition tDCS (HD-tDCS), regardless of the number of electrodes, if stimulation is optimized for focality or intensity (Bikson et al., 2016). We used modified HD transcranial DC stimulation using a small electrode (radius, 0.5 mm). Therefore, we call this “HD-tDCS” in our manuscript. In order to avoid misunderstandings, a part of the Introduction section has been revised.

As for your comment, the electricity induced by HD-tDC stimulation may run via the skin on the back of the neck when an extracephalic reference electrodes is used. However, the use of extracephalic reference electrodes allows for the robust and intuitive estimation of cortical modulations with little dependence on the reference electrode location (Im et al., 2012, PMID 22452936). Moreover, we employed a epicranial electrode implant model (epicranial stimulation of the rat brain, stimulation applied over the skull), and our 3D tDCS simulation results showed that the 63.7-A/m^2^ current flowing from the anode electrode reaches the brain, and that the brain regions close to the anode electrode were intensively stimulated.

8) This work has the potential to be of high translational impact. However, it is not clear how the stimulation intensity used here relates to studies in humans, which is critical for its translational impact, and therefore its suitability for publication in eLife. Can this dosing be scaled to human dosing?

We appreciate this constructive comment. Although we employed a rat epicranial electrode montage that provides for an additional safety margin (Liebetanz et al., 2009; Jackson et al., 2017), the charge density was much higher than that currently applied in human clinical studies. Thus, our results cannot be directly transferred to clinical situations. However, we have to consider head size to transfer stimulation parameters and dose regimes to achieve comparable conditions between animal and human clinical studies. The electric field strengths in a mouse model roughly relate to those in humans at 100:1; a typical transcranial electric stimulation intensity of 1–2 mA in humans approximately translates to 10–20 μA in mice (Alekseichuk et al., 2019). We used an intensity of 50 μA, which results in 5 mA if we convert it into a human dose, the maximum output in tDCS in patients (Furubayashi et al., 2007, PMID 17940759). Moreover, our animal results encourage the development of intensified tDCS protocols to produce more stable or more potent, i.e., therapeutically longer-lasting benefits of neuromodulation in psychiatric disorders (Liebetanz et al., 2009; Jackson et al., 2016; Antal et al., 2017). We now describe this issue in the Discussion section.

9) While some results separate by dose, others do not, and does this question focality? tDCS of M1 showed similar effects to those of the PFC group in both reduction in hyperactivity component (Results paragraph one and Figure 1A) and in passive avoidance test (Figure 1E). This might be due to spillage of current.

Thank you for your constructive comment. This may be a limitation of our study, in that the function of tDCS has not been fully elucidated (please also see our response to comment 6). The mechanisms underpinning tDCS findings are controversial, as studies show that the current reaching the cortex may not be focused enough to entrain neural activity in specific regions according to the location of the reference electrode. Regarding your comment, the application of tDCS over M1 showed similar effects to those observed in the PFC group regarding some behavioral results. This may be explained by a variety of factors, as seen in the response to comment 6 above. We now describe this issue in the Discussion section of our manuscript.

10) The effects of the surgical procedure and the post-operative measures were not investigated as a factor. The stress of the surgical procedure and the restraint required for proper sham or tDCS application and the inconvenience/pain that might be due to the electricity in the real tDCS might be a factor that was not investigated. For example, the timing of the biochemical sampling of TH and BDNF levels is not necessarily related to tDCS beneficial effects. The sampling was 8 days after the last stimulation. Whether this is related to behavior or stress is really not answered. Reviewers had mixed views on if it would be better/needed if the WKY would also contain sham and tDCS, on the expense of the M1 group for example. The positive control group of the MPH is useful.

We appreciate this helpful comment. We performed additional experiments on stress and depression levels, that is, on resting times in the open field test, weight during the experiments, and corticosterone levels in the serum at 2 days after the last tDCS application. We also included additional WKY, WKY-sham, and WKY-tDCS-PFC groups and conducted behavioral tests. There were no significant differences in resting times, weight, or corticosterone levels between the WKY-tDCS and SHR-tDCS groups and naïve or sham animals (Figure 2—figure supplement 1). To clarify the analysis on TH and mBDNF, we performed an additional ELISA analysis using the control strain WKY (WKY, WKY-sham, WKY-tDCS-PFC) and SHR (SHR, SHR-sham, SHR-tDCS-PFC) groups at 2 days after the last application (Figure 2A, B).

[Editors' note: further revisions were suggested prior to acceptance, as described below.]

The manuscript has been improved but there are some remaining issues that need to be addressed.1) The original concern regarding stimulation intensity and focality remains. An essential question that remains unanswered is how the brain electric field distribution in this study compares to typical human tDCS (or what is achievable with human tDCS). While the authors have created a model as recommended, it is not clear how their use of the model addresses this fundamental issue. The model should address 1) whether the stimulation produces too strong of an electric field in the brain to be comparable to human tDCS (intensity) and 2) the degree to which this stimulation is focal (e.g. restricted to PFC or M1). Instead the authors seem to use the modeling results to show that brain current density peaks under the anode and moves when the anode is moved. These two points are mostly trivial (they have been noted by many other studies). Moreover, the way that the model is presented (Figure 6) does not facilitate any quantitative comparison with human tDCS or other animal mechanistic studies. Some suggestions for how to improve the presentation and discussion of the model results.a) Report electric field: the majority of mechanistic or neurophysiological studies of electrical stimulation have controlled for electric field magnitude. The modeling results should be reported in these units to facilitate comparison across systems.b) Peak electric field: We know what peak electric fields in the human cortex during tDCS are (~1 V/m). How does the peak electric field compare in this study? The peak electric field here may be an order of magnitude higher.c) Focality: several studies have suggested that measurable effects on neuronal firing require ~1V/m electric field. The model can help predict how much of the brain crosses this pseudo-threshold in your study. Is it restricted to PFC/M1? Does it reach subcortical structures? It may be that most of the brain in this study is above or near this pseudo-threshold.d) Colormap: Consider using a colormap where electric field magnitude corresponds to color intensity and perhaps an additional figure where the electric field is thresholded above 1 V/m, making the degree of focality obvious.These changes would help facilitate comparison with human tDCS, and help other researchers determine whether these results are likely to apply to their protocol. They may also help interpret your own results (e.g. perhaps PFC and M1 stimulation exhibit similar effects due to direct electrical stimulation of subcortical structures with both montages).

We appreciate the reviewer’s kind and helpful comments. We have now generated new three-dimensional (3D) tDCS simulations of four structures, including cerebrospinal fluid. We have also added information about electric field intensity and current density to facilitate comparisons with human tDCS and other studies. Our 3D model analysis predicted peak electric field intensities of 2.4 V/m and 3.5 V/m in PFC and M1, respectively. The predicted peak current density was 2.1 A/m^2^ in PFC and 3.3 A/m^2^ in M1. These results are described in the Materials and methods section and compared with the rodent and human tDCS results in the Discussion section.

As the reviewer highlights, an electric field as low as ~1 V/m can affect the timing of action potentials (Terzuolo and Bullock, 1956, PMID: 16589932; Liu et al., 2018; Voroslakos et al., 2018). However, higher intensities are required to measurably affect local field potential and associated brain networks, corresponding to intracranial fields of approximately 2 V/m (Liu et al., 2018; Voroslakos et al., 2018). The aims of our study were to identify the therapeutic effects of tDCS for cognitive dysfunction in ADHD and to specifically investigate alterations in dopaminergic neurotransmission and neurotrophic factors at the core sites of mesocorticolimbic dopaminergic pathways. Therefore, additional figures are presented for areas where the electric field is thresholded above 2 V/m to show the degree of focality for PFC and M1. Unfortunately, we could not assess the effects on the activation of subcortical structures due to the limitations of our CoMet program (impossible cross-section view). We have now included a discussion on the higher than expected electric field intensity and current density in our manuscript.

2) With regard to modeling technique, you should explicitly model CSF with a different conductivity from the brain itself. CSF strongly affects current flow patterns due to its high conductivity.

We have generated new three-dimensional tDCS simulations of structures, including cerebrospinal fluid.

3) The western blotting has been removed altogether from the manuscript in response to the comment that BDNF was not in the size that fits the mature form. The authors' claim that they cannot display the western blots according to the journal requirements-the argument for this is not clear. Please explain. And if removed from the results, then reference to them should be removed from the Materials and methods, too.

In the first submission, we included western blot images as a supplementary figure to the corresponding table. However, due to the journal’s formatting restrictions, we could not attach a supplementary figure to the table in the second submission. Thus, we had to exclude the western blot images. In this submission, we have included the western blot images as a supplementary figure to the most relevant figure (Figure 2).

4) The authors depend on ELISA for measuring BDNF. The assay used is actually for a human BDNF. It is true that there is ~50% cross-reactivity, but is there good justification that this is a reliable BDNF assay for the rat?

We measured the amount of BDNF in brain tissue using a commercial ELISA kit. The BDNF gene is conserved in numerous species, including chimpanzees, rhesus monkeys, dogs, cows, mice, rats, chickens, zebrafishes, and frogs (https://www.ncbi.nlm.nih.gov/gene/627). As the commercial kit employs antibodies against human or human recombinant BDNF, it can also be applied to samples from other animals because of the high homology in the protein sequence across species (Klein et al., 2011, PMID: 20604989). As mature BDNF and pro-BDNF elicit opposing biological effects by activating distinct receptors (Yang et al., 2014, PMID: 24746813), the choice of ELISA kit for mature BDNF was particularly important in our study. Lim et al. (2015, PMID: 25824396) reported that the anti-BDNF from the ELISA kit from R&D Systems recognizes mature BDNF with relatively high reactivity compared to the anti-BDNF from other commercial kits. Therefore, we employed the Quantikine BDNF ELISA kit (R&D Systems) for measuring free mature BDNF in our study. The Quantikine BDNF ELISA kit shows < 50% cross-species reactivity and has previously been used for the detection of rat BDNF (Golder et al., 2008, PMID: 18305238; Lim et al., 2015, PMID: 25824396).